# Vitamin E TPGS-Based Nanomedicine, Nanotheranostics, and Targeted Drug Delivery: Past, Present, and Future

**DOI:** 10.3390/pharmaceutics15030722

**Published:** 2023-02-21

**Authors:** Abhishesh Kumar Mehata, Aseem Setia, Ankit Kumar Malik, Rym Hassani, Hamad Ghaleb Dailah, Hassan A. Alhazmi, Ahmed A. Albarraq, Syam Mohan, Madaswamy S. Muthu

**Affiliations:** 1Department of Pharmaceutical Engineering and Technology, Indian Institute of Technology (BHU), Varanasi 221005, India; 2Department of Mathematics, University College AlDarb, Jazan University, Jazan 45142, Saudi Arabia; 3Research and Scientific Studies Unit, College of Nursing, Jazan University, Jazan 45142, Saudi Arabia; 4Substance Abuse and Toxicology Research Centre, Jazan University, Jazan 45142, Saudi Arabia; 5Department of Pharmaceutical Chemistry and Pharmacognosy, College of Pharmacy, Jazan University, Jazan 45142, Saudi Arabia; 6Clinical Pharmacy Department, College of Pharmacy, Jazan University, Jazan 45142, Saudi Arabia; 7School of Health Sciences, University of Petroleum and Energy Studies, Dehradun 248007, India; 8Center for Transdisciplinary Research, Department of Pharmacology, Saveetha Dental College, Saveetha Institute of Medical and Technical science, Saveetha University, Chennai 602105, India

**Keywords:** nanomedicine, TPGS, nanotheranostics, multidrug resistance, clinical trials, patents, targeted drug delivery

## Abstract

It has been seventy years since a water-soluble version of vitamin E called tocophersolan (also known as TPGS) was produced; it was approved by USFDA in 1998 as an inactive ingredient. Drug formulation developers were initially intrigued by its surfactant qualities, and gradually it made its way into the toolkit of pharmaceutical drug delivery. Since then, four drugs with TPGS in their formulation have been approved for sale in the United States and Europe including ibuprofen, tipranavir, amprenavir, and tocophersolan. Improvement and implementation of novel diagnostic and therapeutic techniques for disease are goals of nanomedicine and the succeeding field of nanotheranostics. Specifically, imaging and treating tumors with nanohybrid theranostics shows promising potential. Docetaxel, paclitaxel, and doxorubicin are examples of poorly bioavailable therapeutic agents; hence, much effort is applied for developing TPGS-based nanomedicine, nanotheranostics, and targeted drug delivery systems to increase circulation time and promote the reticular endothelial escape of these drug delivery systems. TPGS has been used in a number of ways for improving drug solubility, bioavailability improvement, and prevention of drug efflux from the targeted cells, which makes it an excellent candidate for therapeutic delivery. Through the downregulation of P-gp expression and modulation of efflux pump activity, TPGS can also mitigate multidrug resistance (MDR). Novel materials such as TPGS-based copolymers are being studied for their potential use in various diseases. In recent clinical trials, TPGS has been utilized in a huge number of Phase I, II, and III studies. Additionally, numerous TPGS-based nanomedicine and nanotheranostic applications are reported in the literature which are in their preclinical stage. However, various randomized or human clinical trials have been underway for TPGS-based drug delivery systems for multiple diseases such as pneumonia, malaria, ocular disease, keratoconus, etc. In this review, we have emphasized in detail the review of the nanotheranostics and targeted drug delivery approaches premised on TPGS. In addition, we have covered various therapeutic systems involving TPGS and its analogs with special references to its patent and clinical trials.

## 1. Introduction

Nanotechnology for the specific imaging and treatment of various illnesses, including cancer, has recently made extensive use of vitamin E TPGS or TPGS [1]. TPGS has been recognized by the FDA as a safe pharmaceutical excipient with the added advantages of its excellent biocompatibility, enhanced drug solubilization, and selectively improved permeability of the anticancer drug across tumor cells [2]. TPGS has demonstrated its key role in inhibiting ATP dependent P-glycoprotein and hence overcoming multidrug resistance during cancer therapy [3]. Combining nanotechnology with TPGS creates a solid foundation for the study and advancement of targeted nanomedicine and nanotheranostics, both of which have great promise for enhancing cancer diagnosis and treatment [4,5]. The nanotheranostics platform with integrated TPGS has depicted promising outcomes in preclinical studies along with enhanced solubility and stability of the loaded drug and diagnostic agents. Numerous studies have suggested that TPGS can avert tumor invasion and metastasis; however, their underlying mechanism remains unexplored [6]. Additionally, the immunological aspects of TPGS require rigorous investigation to develop safe and effective theranostic nanomedicine without any immunological side effects. Currently developed TPGS-based targeted nanomedicine and nanotheranostics are still at the laboratory scale with limited preclinical evaluation; however, progress in developing advanced nanotheranostics remains comparatively slow, hindering its translation into clinical practices [7]. It can be accelerated by optimizing the industrial scale-up process and selecting competent models to eliminate the physiological variation between animals and humans. Further, advancement in nanotheranostics provides a disease prognosis with personalized therapy that enables fatal diseases to be curable or at least treatable at the primary stage [6]. TPGS is a vitamin E derivative that can dissolve in water, and its developmental history goes back 70 years. Its surfactant features caught the attention of drug formulation designers, and it gradually made its way into the toolbox of pharmaceutical drug delivery innovators. Recently, in a study, Liu et al. developed Soluplus^®^ and TPGS-based novel polymeric formulation of curcumin for enhancing its cellular uptake and permeability across the cellular membrane. Additionally, in vitro studies demonstrated that TPGS has successfully improved the water solubility of curcumin and increased its cellular permeability across Caco-2 cells [8]. It was noted that TPGS has antineoplastic action towards breast cancer cells by downregulation of the anti-apoptotic proteins [9]. The surfactant property of the TPGS enables its application in a wide variety of formulation development, including the development of PLGA nanoparticles for DNA labeling, provides uniformly sized nanoparticles with high drug loading, improves their hydrophilicity, and suppresses their tendency to aggregate [10]. Additionally, vitamin E or tocopherols are organic compounds preferentially soluble in lipids with antioxidant properties. Apart from drug delivery, in a recent study, Bi et al. explored incorporation of TPGS in a film of the chitosan along with silicon dioxide nanoparticles, which significantly improved the antioxidant and antimicrobial properties of chitosan film for the packaging [11]. TPGS has been widely used to prepare biocompatible polymer conjugates for drug delivery and targeting [12]. Drug delivery systems for a wide range of ailments, including cancer, tuberculosis, ophthalmic diseases, inflammatory diseases, etc., have been developed using TPGS, demonstrating the technology’s value [13].

The term “theranostics” describes the practice of combining diagnosis and treatment at the same time [14,15,16]. One of the main goals of nanomedicine methods for advanced theranostics is to use and develop nanocarriers like micelles, liposomes, polymeric nanomaterials, quantum dots, etc. The goal is to detect and treat diseases while they are still relatively easy to treat or perhaps completely reverse them if caught early enough. Even for deadly diseases like cancer, cardiovascular disease, and AIDS, nanodiagnostics show promise, offering the possibility of making treatment considerably less bothersome and the prognosis better, reducing healthcare costs, and improving patients’ quality of life [17]. Nanomedicine, for medical purposes, has the potential to be an effective tool in the creation of theranostic candidates, which can be used to diagnose and treat diseases at the same time [18]. Perhaps the most cutting-edge advancement in nanomedicine, the search for combined therapeutic and diagnostic qualities in a single delivery platform is driving the discipline of nanotheranostics. Resveratrol is widely present in grapes and red wine and has been found to have antioxidant and anticancer properties. Although resveratrol’s oral absorption is about 75%, its bioavailability falls below 1% due to its extensive metabolism in the liver and intestine. To overcome such problems, a TPGS and poly-lactide (PLA)-based drug delivery system has been developed for resveratrol [19]. The advanced field of nanomedicine with diagnostic and therapeutic qualities integrated into a single platform is known as nanotheranostics [20]. To transfer several capabilities to a single delivery platform, nanodiagnostics is commonly created using complicated synthetic procedures. To enhance the permeability and retention (EPR) effect and permit the passive accumulation of particles in diseased tissue, nanotheranostics typically employ surface modifications [21].

However, the “responsiveness” of the carriers to external stimulation can provide further selectivity for the diseased area that can be used in nanotheranostics testing. This “activation trigger” is applied non-invasively and remotely to the target location [22]. The vast majority of carriers used for diagnostic or therapeutic purposes are always “on”, with detection and payload release beginning at the point of delivery. Alternatively, carriers having “off–on” theranostic qualities can be designed, which can provide individual assessment of medication delivery to the diseased tissue. Therefore, there may be new opportunities for adjusting the dosage and frequency of treatments made possible by responsive nanotheranostics [23,24]. Under these conditions, it appears difficult to further expand the complexity of these technologies. Numerous current theranostics allow for multimodal imaging and therapy, which improves both diagnostic precision and therapeutic efficacy. However, recent developments in nanomedicine are focusing on making carriers that can adapt to their biological surroundings. In other words, in vivo nanotheranostics can be “triggered” by the physiological changes that distinguish diseased from healthy tissue [25]. These ideas also demonstrate promise for enhancing standard experimental methods. Background noise difficulties associated with imaging of fluorescently modified carriers, for instance, could be alleviated by using nano theranostics that, in accordance with very specific biological inputs, can improve their detection signal [26]. In a recent study, Jasim et al. reviewed the pharmaceutical applications of TPGS, including its surfactant property, synergistic anticancer activity, and induction of programmed cell death in cancer cells. Additionally, they discussed TPGS based prodrugs and their drug delivery systems but were not limited to enhancement of oral bioavailability of the drugs [27]. In this review, we have covered the various developments in the field of TPGS-based drug delivery systems from the past to the present. Additionally, we have discussed novel properties of TPGS, cellular and molecular mechanisms, TPGS-based multifunctional co-polymeric nanomedicine, nanotheranostics, and their chemical modification and conjugation with drugs for improving and advancing the therapeutic properties of the loaded therapeutic agents. Further, we have briefly discussed the pharmacokinetic and the safety assessment of TPGS to be used as suitable carriers for drug delivery but not limited to their role in the enhancement of the diagnostic and therapeutic properties of TPGS-based nanotheranostics systems. Finally, we have limited our discussion to clinical trials, regulatory status, and TPGS-based marketed formulations for numerous diseases.

## 2. Historical Milestones Achieved in TPGS Developments

Eastman Kodak Company, in the year 1950, invented the novel polymer named Vit. E TPGS, a water-soluble derivative of natural Vit. E as a supplement in Vit. E deficiency. TPGS is prepared by esterification of the natural Vit. E and PEG 1000. TPGS has been known to demonstrate higher water solubility compared with its parent molecules [28]. The solubility of TPGs was noted to be 20% *w*/*w* at 25 °C. Ten years after its invention (1960), TPGS was recommended as a solubilizing agent for water-insoluble vitamins. In the 1970s, the toxicology and safety profile of TPGS was published, demonstrating that TPGS is safe to be used [29]. In the 1980s, TPGS was shown to play a significant role in the treatment of Vit. E deficiency in cholestatic children and zoo animals [30]. Later, its effectiveness as a hydrophilic antioxidant was reported, but this activity was observed only after hydrolysis of TPGS. The majority of TPGS characteristic application was reported in the 1990s. The enhancement in the oral absorption of cyclosporine and vitamin D was presented in 1996 [31]. Additionally, TPGS liquid crystalline phase properties, enhanced cellular uptake of protease inhibitors, and carrier capability for hemostat, have been reported in the early 1990s. Further, TPGS was approved as a solubilizer and absorption enhancer in pharmaceutical formulations in 1999. Furthermore, the first FDA-approved formulation of amprenavir containing TPGS was reported in 1999 [32]. Numerous studies reported the usage of the TPGS and its analogs in the absorption enhancement of poorly absorbed drugs in the earlier 2000s. TPGS has recently received a lot of attention for its pharmaceutical innovation research. The proportion of publications and patents in the last 20 years precisely illustrates this pattern [33].

## 3. TPGS Characteristics and Synthesis

### 3.1. Novel Properties of TPGS That Enable It to Be Used as a Promising Excipient in Nanomedicine and Targeted Drug Delivery Systems

TPGS is an amphiphilic compound with a PEG chain as a hydrophilic head and Vit. E as the lipophilic tail that reflects nonionic surfactant properties. TPGS has an HLB value of 13 [34]. Consequently, TPGS has the property to enhance the solubility of a lipophilic compound in an aqueous environment through the process of emulsification. Alteration in the PEG chain length in the TPGS alters its HLB number. For instance, TPGS comprised of PEG 200, PEG 1000, and PEG 2000 has HLB numbers of 9, 13, and 16, respectively [35]. The commercial form of TPGS has PEG 1000 in its structural skeleton. TPGS appears as a waxy solid material having a melting point of 37 °C. The viscosity of TPGS decreases with an increase in temperature [36]. For example, TPGS has 400 cps viscosity at 50 °C and 230 cps viscosity at 60 °C. TPGS has been shown to be sufficiently stable to withstand heating and cooling cycle modelling. Hence, TPGS can be used as a promising excipient in the development and advancement of pharmaceutical formulations. The uptake improvement features of TPGS were initially documented in 1992, and the improving effect of TPGS on vitamin D was explained [33]. Enhanced absorption of an anti-HIV protease inhibitor was recommended in 1999 as a result of the drug’s enhanced solubility and permeability; utilizing Caco-2 cell monolayers, the suppression of the P-glycoprotein (P-gp) efflux pump by TPGS with numerous PEG chain sizes were evaluated. The side chain of PEG 1000 was found to be close to ideal [37].

To produce TPGS, ɑ-tocopheryl succinate (ɑ-TOS) is esterified with PEG1000. This amphiphilic agent’s favourable HLB value makes it effective as a solubilizer, emulsifier [38], penetration and absorption-enhancer of hydrophobic medications. Increased medication efficacy against multidrug-resistant (MDR) cells, decreased particle size, and greater solubility are all possible outcomes of TPGS-formed micelles in an aqueous solution [39]. By inhibiting P-gp expression and modifying efflux pump activity, TPGS can also reduce MDR [40]. Therefore, many TPGS-based nano-delivery methods have been studied to counteract P-gp-mediated multidrug resistance [41]. Jiang et al. [42] created TPGS-laponite nanodisks, a novel pH-responsive hybrid DDS to combat MDR. DOX-loaded TPGS-modified laponite nanodisks showed remarkable tumor suppression of MCF-7/ADR with minimal toxicity in vitro and in vivo studies. For the preparation of temoporfin-loaded RGD-nanoparticles, Wu et al. [43] used vitamin-E-succinate-grafted chitosan oligosaccharide and TPGS. This strategy combines chemotherapeutic and photodynamic approaches. The therapeutic efficacy of the particles was significantly increased by their ability to selectively target cancers high in integrin [44]. To maximize the efficacy of photochemotherapy, the matrix material was developed to be biocompatible, have a high loading capacity for chemotherapeutics, be very stable in biological fluids, and inhibit P-gp. Nano assemblies’ targeting and penetration efficiency were enhanced by using a peptide that recognizes the tumor vasculature and a peptide that penetrates the tumor, called iNGR. Nano assemblies utilizing the aforementioned methods have a potent antitumor effect in vitro and in vivo, even against drug-resistant tumor cells [45]. Combining chemotherapy with reduced response and photothermal therapy increases tumor penetration [44].

### 3.2. Key Role of Vitamin E and Its Derivatives for the Protection of Body Cells

Vitamin E is an essential component of cells that protects them from being damaged due to oxidative stress because of free radical generation and accumulation inside the cells. It protects the membranes of erythrocytes and lung cells from the damaging effects of free radicals due to its antioxidant properties. Further, it has anti-inflammatory properties due to its ability to suppress the protein lipoxygenase [46]. Human tissue and plasma are rich in alpha-tocopherol (α-TOS), one of eight isoforms of vitamin E (all of which are lipid-soluble). The Vit-E chromanol head and phytyl tail make up its structure (Figure 1). As a means of increasing its anticancer efficacy and biocompatibility, several structurally modified analogs of α-TOS have been produced. It has been reported that TPGS is not toxic and does not produce any adverse events after administration at the dose of 1000 mg/kg body weight/day. Furthermore, TPGS’s anticancer activity was found to be superior to α-TOS, emphasizing its significant application as an adjuvant in the development of nanoparticulate drug delivery systems [47].

### 3.3. TPGS in the Role of Solubilization Promoter

The drug delivery system makes extensive use of TPGS because of its solubilization feature, which allows it to transport drugs that are extremely lipid-soluble. TPGS can be easily diluted with any liquid, including oils, surfactants, and water. Human hepatocellular carcinoma cells exhibited higher apoptosis when tanshinone IIA was administered using the TPGS micellar system. The addition of camptothecin to P105/TPGS mixed micelles enhanced their solubility and cytotoxicity in a similar fashion. Incorporating the weakly soluble medication formononetin into phospholipid/TPGS micelles strengthened its anticancer impact because of enhanced cellular absorption and cytotoxicity [48]. A similar strategy was used to load (docetaxel) DTX into PEG/TPGS for the treatment of MDR cancer [49]. Increased cytotoxicity and better cellular absorption of DTX were seen in drug-resistant H460/TaxR cancer cells after the addition of TPGS to the DTX-loaded formulation. TPGS is used as a solubilizer in many commercial products today, including BioResponse-DIM, Agenerase, Nurofen, and Wal-profen. Despite efforts to make the chemotherapy drug paclitaxel more water-soluble, the Torosol emulsion failed in the phase III clinical study because it was not selective enough for cancer cells [47].

### 3.4. TPGS as a P-gp Modulator with Anticancer Properties

TPGS anticancer effects have been found both when used alone and in combination with standard chemotherapies. When comparing TPGS and α-TOS, the latter was found to be more effective at suppressing growth. The high efficiency with which TPGS generates reactive oxygen species (ROS) and, consequently, promotes apoptosis may account for its potent anticancer activity against H460 cells. TPGS anticancer action is mediated by PEG conjugation. TPGS was found to generate more ROS, induce more apoptosis (cell death), and limit growth compared to TOS [50]. TPGS can trigger apoptosis in two different ways: with and without the help of the death-inducing caspases. Mechanism analysis showed that TPGS inhibits AKT phosphorylation, which in turn causes anti-apoptotic proteins like Survivin and Bcl-2 to be downregulated and eventually apoptosis to occur (Figure 1) [51]. In addition to having an anticancer impact, TPGS also has the additional advantage of blocking P-gp efflux, making it a more effective cancer treatment. Cancer cells become resistant to treatment when their overexpression of P-gp causes an increase in the efflux of harmful chemicals from the cell. TPGS blocks P-gp efflux by inhibiting P-gp ATPase, the enzyme responsible for providing the energy for P-gp transport. Additionally, TPGS causes the cell membrane to stiffen, which in turn decreases P-gp transport from the basolateral to the apical membrane. Neither a substrate nor a competitive inhibitor, TPGS is ignored by the P-gp efflux pump. A TPGS chain length of 1000 Da was found to be maximally inhibitive of the P-gp efflux pump, while a chain length of 1500 Da was necessary for optimum suppression [52].

MDR stands as a stumbling block in the way of effective clinical treatment. The several MDR mechanisms are complex. Overexpression of P-gp is a contributing factor. P-gp inhibitors have received a lot of attention in the past few decades as a potential method of combating multidrug resistance [53]. Pluronic, Tweens, Span, and TPGS are just some of the many nonionic surfactants that have been shown to suppress P-gp activity. Among these surfactants, TPGS is the most effective P-gp inhibitor currently on the market. Below the critical micelle concentration (CMC) of 0.02 wt%, TPGS acts as an inhibitor of P-gp by blocking the drug efflux pump [54]. Rege et al. [55] observed the influence of TPGS on membrane fluidity. This study disproved the hypothesis that alterations in membrane fluidity are a universal mechanism for decreasing transporter activity. For their series of research on the mechanism of P-gp suppression by TPGS, Collnot et al. [56] advocated utilizing the Caco-2 cell line. Based on the results, TPGS did not function as a platform or competitive inhibitor of P-gp efflux transport. Furthermore, TPGS did not have any appreciable effects on the ATP level, ruling out any interference with mitochondrial activity or depletion of intracellular ATP. Hao et al. [57] showed that TPGS can decrease ATP utilization and membrane potential in mitochondria, suggesting that TPGS can alter mitochondrial function in addition to reducing ATPase activity to inhibit P-gp. Additionally, Wang et al. [58] demonstrated that TPGS might inhibit P-gp activity by inducing mitochondrial apoptosis in (DTX)-loaded DSPE/PEG hybrid micelles (TPGS/DTX-M) and folic acid-decorated TPGS/DTX-M (FA@TPGS/DTX-M). Zhu et al. [59], in a Western blot experiment, also suggested that TPGS inhibited P-gp expression, therefore reversing MDR.

### 3.5. Chemical Modification of TPGS

TPGS most often has a molecular weight of 1513 g/mol, a critical micelle concentration (CMC) of 0.02 percent weight-wise, and an HLB value of 13.2 [6]. Chemical groups can also be added to TPGS to alter their properties so that necessary functional groups can be used to bind ligands over nanomedicine surfaces. Numerous studies used TPGS–COOH to prepare COOH functionalized nanomedicines that can be post-conjugated with monoclonal antibodies that are specific to certain receptors, such as cetuximab for EGFR targeting and bevacizumab for VEGF [60]. In a study, Viswanadh et al. prepared TPGS-SH by reacting TPGS–COOH with 4-aminothiophenol in the presence of carbodiimide crosslinkers (EDC/NHS). Prepared TPGS-SH was used to synthesized redox-sensitive nanoparticles [61]. In another study, Li et al. prepared TPGS-NH_2_ by reacting with triethylamine in the presence of p-toluenesulfonyl chloride (used as an activator for primary alcohol). The resulting TPGS-NH_2_ was then conjugated with folic acid in the presence of EDC/NHS to produce TPGS-Folic acid [62].

### 3.6. Drug–TPGS Conjugate

Various drug–TPGS conjugates were prepared by different researchers to enhance their therapeutic effects. Cao et al. prepared TPGS–doxorubicin conjugate in a two-step reaction. The first step includes the preparation of TPGS–COOH as discussed previously. In the second step, doxorubicin was reacted with TPGS–COOH by employing DCC and NHS. The TPGS–doxorubicin conjugate showed improved cytotoxicity, more potent against MCF-7 cells, as compared to doxorubicin control. The IC_50_ value of drug–TPGS conjugate was 1.5-fold less than that of the free drug after 24 h. Moreover, the pharmacokinetic study was performed on Sprague–Dawley rats. The doxorubicin–TPGS conjugate exhibited 4.5 folds higher half-life and 24 folds higher AUC (area under the curve) than the doxorubicin control when administered intravenously at the dose of 5 mg/kg [63].

Similarly, Bao et al. prepared the conjugate of TPGS–paclitaxel that can be self-assembled to form colloidal nanocarriers. First, they prepared TPGS–SS–COOH (dithiodipropionic TPGS ester), which was then conjugated with the paclitaxel using DCC/NHS (Figure 2). The prepared prodrug was self-aggregated in physiological media. The self-aggregated nanocarriers of the TPGS-paclitaxel conjugate exhibited improved cytotoxicity in paclitaxel-resistant human ovarian cell lines (A2780/T). When compared to drug control, the in vivo evaluation of the TPGS–paclitaxel conjugate showed a longer half-life, increased AUC, and a substantial reduction of tumor growth [64].

Mi et al. developed a TPGS–cisplatin conjugate system to form a micelle system. When compared to the control drug, the IC_50_ value for HepG2 hepatocarcinoma cells was reduced by almost three times in the cell viability assay, which proved the significant enhancement of the cisplatin chemotherapy [65].

## 4. Pharmacokinetic and Safety Assessment of TPGS

Since TPGS has an acute lethal dosage (LD_50_) of more than 7 g/kg for adult rats, it is utilized in a broad variety of pharmaceutical products and is regarded as safe [6]. Zang et al. studied the pharmacokinetics of TPGS, including its absorption, distribution, metabolism, and excretion, after being administered to rats orally and intravenously. According to the pharmacokinetic analysis, the bioavailability of TPGS following oral treatment in rats remained extremely low, whereas, after intravenous injection, TPGS and PEG1000 were present in the plasma samples. Following intravenous injection, the plasma concentration–time profiles of TPGS showed that it was quickly distributed to tissues and organs. The larger volume of distribution (V_d_) of TPGS indicates that it’s in vivo biodistribution is high. The elimination half-life (T_1/2_) of TPGS is as high as 11 hrs. The presence of PEG1000 in the plasma samples is caused by the rapid hydrolysis of the TPGS after intravenous injection. PEG1000 is rapidly excreted in comparison to TPGS, which minimizes the chances of it accumulating in tissues. The spleen, liver, and lungs have higher concentrations of TPGS, which correlate to organs with high blood perfusion rates and high reticuloendothelial system expression (RES). Due to the very high blood perfusion rates in the heart and kidney, a low expression of the RES in these tissues likely explains why the quantities of TPGS in these tissues are so much lower. The fact that TPGS cannot pass the blood–brain barrier may explain why there is so little of it in brain tissue. Since muscle and fat have poor blood perfusion rates, their tissue levels are also low. It is interesting to know that while the concentration of TPGS in the ovaries is higher than that in the heart and kidney and lower than the lower limit of quantitation (50 ng/mL), the quantity in the testes is below that level. After intravenous injection, TPGS was undetectable in urine, demonstrating that renal excretion is not the route by which it is cleared; nevertheless, PEG1000 may be eliminated in faeces and urine. TPGS might be hydrolyzed to PEG1000 by the carboxyl esterase 1 enzyme in cells, but it is not likely to be hydrolyzed by pancreatic lipases in the digestive system (Figure 3). Although pancreatic lipases in the digestive tract are not likely to hydrolyze TPGS, the carboxyl esterase 1 enzyme in cells may hydrolyze it to PEG1000. TPGS only weakly inhibits CYP450 enzymes, with the exception of CYP3A4, which may be moderately inhibited. If this is a clinical risk, TPGS should be determined by the plasma concentration obtained during clinical applications of a TPGS-based formulation (Table 1). Since TPGS is frequently present in significant amounts in these nanomedicines, it would seem at least conceivable that a concentration of TPGS high enough to inhibit CYP3A4 may be obtained. This shows that monitoring the plasma levels of TPGS while a TPGS-based nanomedicine is being used therapeutically might be an effective strategy to prevent this form of drug interaction [6].

## 5. Multifunctional TPGS-Copolymer-Based Nanomedicine

Safe, stealthy, targeting feasible, multifunctional, biocompatible, tunable in size, and simple to prepare, copolymers have been employed in the manufacture of nanoformulations. As a rule, the copolymer is made up of repeating units of two different monomers [70]. Types of random copolymers include alternative copolymers, block copolymers, and graft copolymers. Graft copolymers have monomer chains linked in various locations on host polymers, while random copolymers have repeating units arranged in a disorderly fashion. They have a systematic arrangement in the case of alternative copolymers [71]. There can be no grafting process without reactive functional groups on the host polymer’s structure. The new systems of targeted nanomedicines aim to increase the drug’s specificity for the tumor site, decrease its toxicity, and increase its effectiveness. Novel materials such as copolymers are being studied for their potential use in the development of both broad-spectrum and cancer-specific nanomedicines [72]. Polymers such as TPGS, PLA, PLA-TPGS, and PLA-PEG [69] have shown their promising potential to be used as a nanocarrier for future use in the development of multifunctional nanomedicine [73]. Active targeting has the ability to be a useful strategy because of the need for various nanoparticles by tumors, which can boost the efficacy of the treatment and decrease the adverse chemotherapy side effects. Surface modification with targeting probes enabled the efficient application of PLA–TPGS nanoparticles for targeted tumor therapy [74]. The folate receptor is abundantly expressed on the cell surface of numerous hematological malignancies, such as those in the central nervous system, renal, breast, ovaries, and respiratory. There is a 100–300-fold increase in folate receptor expression in tumor cells compared to healthy ones [75]. Vitamin B9, or folic acid, plays a pivotal role in the development of cancer cells by aiding in the production of nucleotide bases [76]. In terms of therapeutic efficacy, nanoparticles coated with folate showed markedly enhanced targeted delivery into cancer cells (Figure 4I). The absorption of nanoparticles by cells was increased by a factor of 1.5 when folate was added to the nanoparticles. Green fluorescence from coumarin-6 around the nucleus is much more noticeable for folate-decorated nanoparticles than for bare nanoparticles without a targeting effect [77]. Recently, liposomes surface-modified with TPGS were prepared using docetaxel as a model drug, and their efficacy against brain tumors was compared to that of PEGylated liposomes, conventional nude liposomes (without TPGS coating), and a commercially available formulation of docetaxel (Taxotere^®^) in in vitro cell line studies (Figure 4II). The IC_50_ value demonstrated that the TPGS-coated liposomes were seven times more effective than Taxotere^®^. After 24 h in culture, C6 glioma cells revealed an IC_50_ of 5.93 ± 0.57 μg/mL for TPGS liposomes, 7.70 ± 0.22 for PEGylated liposomes, 31.04 ± 0.75 for nude liposomes, and 37.04 ± 1.05 for Taxotere^®^. In addition, TPGS liposomes were found to have superior internalization compared to the other formulations in investigations of cell uptake utilizing coumarin-6 on C6 glioma cells. Liposomes containing TPGS were found to be more effective and to be taken up by cells at a higher rate than those without TPGS due to the passive transport through the EPR effect and suppression of P-gp-mediated drug efflux by TPGS [78].

In a study, Muthu et al. formulated multifunctional liposomes encasing docetaxel and quantum dots (QDs) coated with TPGS-containing and without targeting ligands (Figure 4III). Overexpressing folate receptors in MCF-7 cell lines were the focus of targeted drug therapy using folic acid as a probe. Cellular absorption and drug cytotoxicity and QDs-loaded liposomes were evaluated for in vitro cell line tests. CLSM pictures taken from MCF-7 cells after 2 h of incubation revealed that the targeting liposomes were taken up more quickly by the cells than the non-targeting ones [9]. Furthermore, Mehata et al. prepared nanoparticles of trastuzumab-decorated or undecorated conjugated chitosan (TPGS-g-chitosan). Entrapment efficiency was between 74% and 78%, and particle size ranged from 126.1 to 186 nm for both conventional and tailored nanoparticles. Using SK-BR-3 cells in vitro, the mucoadhesive characteristic of TPGS-g-chitosan nanoparticles was shown to be promising, while the nanoparticle’s non-targeted and HER-2 receptor-targeted cytotoxicity and cellular uptake were both enhanced. The IC_50_ value of non-targeted nanoparticles was 43 times higher than Docel^®^, while the IC_50_ value of targeted nanoparticles was 223 times higher than Docel^®^. Histopathology studies compared Docel^®^ to those using non-targeted and targeted nanoparticles and found that the latter was less hazardous to important organs such as the lungs, liver, and kidneys (Figure 4IV) [66]. P-gp efflux pump modulators, such as certain functional excipients, have recently attracted the attention of scientists. Ingredients such as Cremophor EL, polysorbate 80, Pluronic P85, and TPGS (Figure 4V) are known as excipients [79]. And perhaps most crucially, PLGA can regulate P-gp drug efflux pumps. In contrast, when PLGA is utilized alone as an nDDS, it has some limitations, including a limited drug loading capacity and the rapid release of pharmaceuticals from transporters. Another promising biomaterial for nDDS development is TPGS. TPGS has a hydrophilic tail with a relatively large lipophilic nonpolar head, suggesting it is a surfactant with excellent emulsification efficiency. This combination may improve drug solubilization [80]. Wang et al. developed TPGS/PLGA/SN-38 NPs. They propose novel pathways of MDR reversal for TPGS/PLGA NPs based on an analysis of intracellular accumulation in relation to time-dependent uptake and efflux inhibition. First, intact NPs made of TPGS/PLGA/SN-38 increased the uptake of the loaded medication via clathrin-mediated endocytosis. At the same time, NPs inside the cell were able to avoid being recognized by P-gp in MDR cells. The efflux surroundings of the P-gp pump, including mitochondria and the P-gp domain with an ATP-binding site, may be modulated by TPGS or/and PLGA after SN-38 has been released from TPGS/PLGA/SN-38 NPs in MDR cells. The cytotoxic effect of the controlled-release medication was ultimately brought about by its entry into the nucleus of the MDR cell (Figure 4V) [81].

**Figure 4 pharmaceutics-15-00722-f004:**
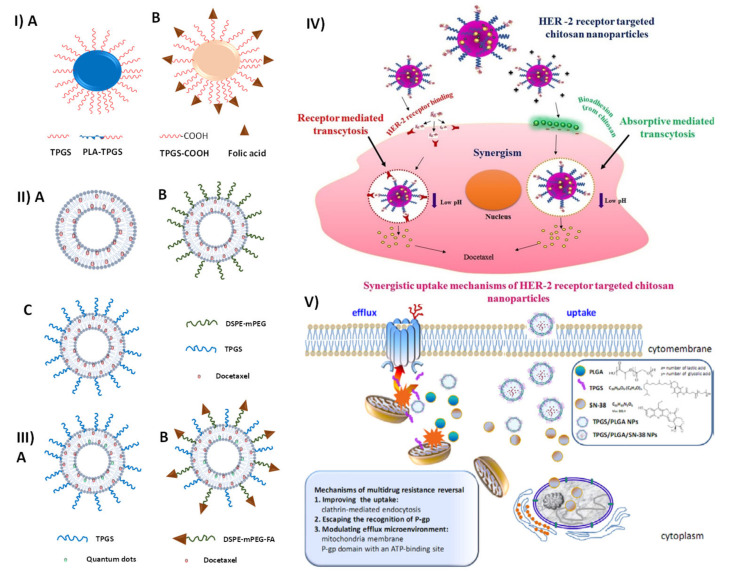
(**I**) (**A**) TPGS-emulsified PLA-TPGS-NPs without a specific target; (**B**) Folate-targeted PLA-TPGS nanoparticle emulsion; (**II**) (**A**) Traditional liposomes, (**B**) Liposomal PEGylation (**C**) Covered lipid vesicles containing TPGS; (**III**) (**A**) nonspecific multipurpose liposomes, (**B**) Liposomes with several targets, including the folate receptor; (**IV**) TPGS-g-chitosan nanoparticles coated with trastuzumab for selective breast cancer therapy; (**V**) Schematic depiction of the novel NPs reversing MDR processes in cancer cells, together with the chemical structures of PLGA, TPGS, and SN-38. (**IV**) Reproduced with permission from Ref. [66], Graphical abstract. (Elsevier 2019). (**V**) Reproduced with permission from Ref. [81], Figure 1 (IOP 2012).

## 6. TPGS-Based Nanomedicine and Nanotheranostics

The amphiphilic nature of TPGS with excellent biocompatibility demonstrates a key driving force in the area of drug delivery research and development. In an aqueous matrix system, TPGS can significantly boost the solubility of an active pharmaceutical ingredient. As evident from numerous published pieces of research, TPGS is a novel polymer that has been used in a wide range of applications in nanomedicine as an emulsifying agent, solubility booster, and permeation enhancer, and in conjugation with other polymers and ligand (such as monoclonal antibodies, and peptides) for improving biological performance and targeted drug delivery, respectively [3]. Over the span of 50 years, after the invention of TPGS, most of the research was based on the applications of TPGS in solubility enhancement, antioxidant agents, and Vit. E deficiency supplements [82]. The major breakthrough in the development of TPGS nanomedicine was achieved in the early 2000s. The bulk of the effort went into perfecting the TPGS-based nanosystems to increase the accessibility of medicines with low oral bioavailability [83], such as docetaxel, paclitaxel, and doxorubicin, and increase circulation time, promoting reticular endothelial escape of nanomedicine [84]. In the 2010s, numerous TPGS-based nanomedicine platforms were reported, including receptor-targeted nanomedicine and nanotheranostics for simultaneous diagnosis and targeted delivery of diagnostic agents and drug molecules. A detailed description of TPGS-related studies is highlighted in Table 2. TPGS has also been used to prepare prodrugs in which TPGS is conjugated with drugs for improving the pharmacokinetic profile of drug molecules. It was noted that doxorubicin (DOX) has a shorter half-life and produces cardiovascular toxicity apart from its anticancer activity. In a study, Anbharasi et al. prepared non-targeted (TPGS-DOX) and targeted (TPGS-DOX-FOL) prodrugs for evaluation in folate receptor overexpressed breast cancer. It was observed that TPGS conjugation in TPGS-DOX and TPGS-DOX-FOL prodrug boosted in vitro cellular uptake, and cytotoxicity significantly, and in vivo experiments demonstrated that the half-life of the prodrug was increased by 3.8-fold in comparison with free DOX [85].

## 7. TPGS-Based Nanomaterials for Theranostics/Targeted Application

### 7.1. TPGS-Based Micelles for Nanotheranostics

The utilization of micellar solutions of amphiphiles in drug delivery is a successful strategy for reaching therapeutic targets. Due to the hydrophobic environment of micelles, it is possible to solubilize water-insoluble medications and then load them for delivery to the desired sites [90]. Targeted drug delivery systems are made to keep drugs from breaking down or getting lost, to reduce serious side effects, and to make drugs more bioavailable. In contrast to ionic dispersion, nonionic micellar dispersions begin to get cloudy at a lower temperature. As a direct result of the detection of larger particles, the clouding phenomenon occurs [94,95]. CMC can be measured in many different ways, such as by interfacial tension, conductivity, osmotic pressure, and more. Polymeric micelles have been developed by considering that formulations might reach therapeutic drug concentrations when administered systemically [96]. The polymeric micelle’s drug-loading efficiency is influenced by the compatibility of polymers and medicines. Encapsulation of drugs is largely dependent on how hydrophobic they are in comparison to the micellar core. By chemical conjugation or by physical trapping by dialysis or emulsification, insoluble medicines can be incorporated into the micellar core [97]. Polymeric micelles, as well as other drug carriers, might be useful in the pharmaceutical industry if drug targeting mechanisms can be better understood. The micelles are able to enter the interstitial space due to a leaky vasculature that allows them to do so. Using micelles made of polymers, the efficacy of medicine may be increased by targeting certain cells and organs without building up in healthy tissues [98].

TPGS-based micellar formulation is a promising nanomedicine platform for imaging applications. In the literature, numerous TPGS-based micellar formulations have been reported, loaded with docetaxel, paclitaxel for transferrin, folate, HER-2 receptor-targeted drug delivery, etc. In a study, Muthu et al. formulated theranostic TPGS docetaxel micelles and transferrin was used for targeted delivery and ultrabright gold nanoclusters (diagnostic agent) were used for breast cancer. In this study, DTX was employed as a reference medication, and ultrabright gold nanoclusters (AuNC) were employed as a paradigm imaging agent. This research aimed to develop TPGS-coupled transferrin micelles for use as a cutting-edge theranostic tool. Using a solvent casting technique, they produced imaging and therapeutic micelles with and without conjugation and described them in a variety of ways. Transferrin receptors expressed in MDA-MB-231-luc breast cancer cells and NIH-3T3 fibroblast cells were utilized to evaluate the cytotoxicity of the formulations in vitro. Using flow cytometry, they found that the transferrin receptor was really overexpressed on the surface of MDA-MB-231-luc cells. Micelles’ biodistribution and theranostic efficacy were examined in SCID mice with MDA-MB-231-luc tumors using the Xenogen IVIS^®^, which enables simultaneous fluorescence and luciferase bioluminescence imaging. After 24 h of therapy, MDA-MB-231-luc cells showed Taxotere^®^ to be 15.31 and 71.73 times more effective than the non-targeted and targeted micelles, respectively. Figure 5 demonstrated the confocal images of MDA-MB-231-luc treated with free AuNC, targeted (DTX-AuTfM), and non-targeted (DTX-AuM) micelles. DTX-AuTfM theranostic micelles (Row 3, top) in MDA-MB-231-luc cells exhibited stronger red fluorescence and more cytoplasmic red stain than AuNC and DTX-AuM micelles (Rows 1 and 2, respectively) (Figure 5I). This finding indicates that MDA-MB-231-luc breast cancer cells incorporated targeted micelles more deeply into their cytoplasm. Competitive inhibition of the absorption of DTX-AuTfM occurs when an excess of added free transferrin inhibits the transferrin receptors. As a result of transferrin receptor inhibition, the red fluorescence intensity and the red stain in the cytoplasm were reduced in Row 4 for DTX-AuTfM. The findings verify the theranostic micelle’s cellular absorption via the transferrin receptor (DTX-AuTfM). After intravenous administration, i.v. theranostic imaging was performed on the control AuNC and fluorescent AuNC formulated in DTX-AuM and DTX-AuTfM theranostic micelles. Within 24 h of the fourth injection on day 24, the fluorescence intensity of the AuNC was drastically reduced everywhere but not in the tumor, the liver, and the urinary bladder Figure 5II(A). The findings indicated that 18 and 42 folds more AuNC formed in non-targeted and targeted micelles, respectively, could be accumulated than the free AuNC, 24 h following therapy with 4th injections within 24 days Figure 5II(A). In comparison to non-targeted micelles, DTX-AuTfM has a higher blood circulation time and accumulated more in the tumor due to the transferrin targeting effects, as indicated by a significant increase in fluorescence intensity. After 12 days of treatment for tumor growth, using AuNC theranostic imaging, the bioluminescent imaging of the Taxotere^®^, AuNC control, DTX-AuM, and DTX-AuTfM theranostic micelles, as well as the control saline, were compared. In this experiment, the bioluminescent intensity of the saline control and the AuNC control did not differ significantly. At the same time, Taxotere^®^ revealed a modest drop in ROI data compared to the saline control. After 12 days of therapy, both the non-targeted and targeted micelles of DTX demonstrated significant tumor growth suppression compared to Taxotere^®^ Figure 5II(B). In addition, DTX-AuTfM showed a statistically significant reduction in ROI data/tumor growth compared to the DTX-AuM Figure 5II(B). This outcome originates from the DTX-AuTfM micelles being purposefully directed in vivo. Utilizing the overexpressed transferrin receptor found on the surface of cancer cells, active tumor targeting with transferrin significantly slowed the progression of the malignancy [86].

In a study, Agarwal et al. prepared targeted bioadhesive TPGS–DTX micelles for brain therapy. In order to characterize the micelles, they used the solvent casting technique to create them. Compared to Docel^®^ (commercial formulation), TPGS–chitosan micelles that target the transferrin receptor were 248 times more effective in eradicating C6 cells after 24 h of treatment, as measured by the IC50 values. Targeted TPGS–chitosan micelles rely on the bioadhesive property of chitosan and transferrin receptor-mediated endocytosis to exert their effects, as demonstrated by studies examining the cellular uptake of the micelles over time. Following 48 h of treatment, targeted and non-targeted micelles were 2.89 and 4.08 times more efficacious than Docel^®^, respectively, according to in vivo pharmacokinetic analyses [67].

Furthermore, Muthu et al. formulated TPGS–docetaxel micelles for brain cancer therapy via the utilization of polyethylene glycol. Docetaxel-loaded TPGS micelles had a particle size of around 12–14 nm. A quantitative metric, IC_50_, was determined to show the benefits of the TPGS micellar formulation of DTX. Each of the three TPGS micelles, DTXSC-50, DTXSC-100, and DTXSC-150, showed to be 53.3, 64.2, and 66.6% more effective than Taxotere after 24 h of incubation, with respective IC_50_ values of 9.31 ± 0.86, 7.13 ± 0.62, and 6.65 ± 0.34 versus 19.92 ± 2.13 µg/mL. To improve drug encapsulation effectiveness, cellular absorption, cytotoxicity, and the desired biodistribution of the produced medicament, the TPGS has a critical micelle concentration that is much lower than that of most phospholipids in the micellar formulation. This suggests that TPGS may be useful as a medication carrier that can pass the BBB [87].

In another investigation, Muthu et al. developed paliperidone palmitate-loaded-TPGS micelles for the treatment of psychosis. The formulation was made with solvent casting, while the control paliperidone palmitate preparations were made with simple sonication. Variables were tested in comparisons between the produced micelles and a control formulation of paliperidone palmitate. Paliperidone palmitate-loaded TPGS micelles had average particle diameters of 26.5 nm. Micelles were effective in medication encapsulation, around 92%. More than 24 h after loading TPGS micelles with paliperidone palmitate, 40% of the drug was still being released from the micelles [90].

For the first time in the literature, Viswanadh et al. synthesized novel glutathione redox-sensitive thiolated TPGS (TPGS-SH). Additionally, cetuximab functionalized Cross-verified TPGS-SH nanoparticles were prepared for EGFR receptor-targeted and redox-sensitive drug delivery in lung cancer. Further, in vitro release study demonstrated that the presence of glutathione in the release medium accelerated docetaxel release. Generally, a higher level of glutathione in the cancer cells (bone marrow, breast, and lung cancers) is responsible for the development of resistance to a number of anticancer drugs. Furthermore, in vitro anticancer activity against the A549 cell line depicted that targeted TPGS-SH NPs have a higher cytotoxic effect and improved cellular uptake. Hence, developed TPGS-SH NPs have promising outcomes in lung cancer with elevated levels of glutathione and resistance to chemotherapeutic agents. The formulation of redox-sensitive nanoparticles is shown in Figure 6I. The amount of wound healing was used to determine the cytotoxic property of redox-sensitive NPs, and the corresponding brightfield pictures are shown in Figure 6II. In the saline control wells, the scrape is nearly fully covered by cells after 48 h Figure 6II. However, in wells fed with redox-sensitive NPs, cell mobility was not detected in the scratch zone, and the scratch area was actually enlarged, most likely due to cell death. The severity of apoptosis generated by DTX-loaded redox-sensitive NPs and DTX control was evaluated in this work by observing morphological changes in cells after treatment. DAPI staining and counterstaining with F-actin were performed on A549 cells that had been incubated with formulations for 6 h Figure 6III [61].

Recently, Bernabeu et al. have developed methotrexate-loaded mixed nanomicelles of sodium deoxycholate and TPGS for improving the aqueous solubility and anticancer activity. In vitro cellular viability study demonstrated that micellar formulations have a lower IC_50_ compared to free methotrexate in mCF-7 and MDA-MB-231 cells. Additionally, the micellar formulation was found to have higher cellular uptake compared to the free drug in both cell lines. Hence, the developed mixed nanomicelles are able to overcome the resistance developed by breast cancer cells and improve their anticancer efficacy [99].

### 7.2. TPGS-Based Liposomes for Nanotheranostics

It was suggested that liposomes coated with TPGS greatly improved the biological performance of liposomes, including enhancing stability, cellular uptake, circulation time, escaping and macrophage capturing, etc. By delaying medication clearance, slowing drug metabolism, reducing distribution volume, and shifting the distribution to cancerous tissues with higher capillary permeability, liposomal drug delivery methods significantly modify the biodistribution of the pharmaceuticals they are designed to deliver [100]. The amount of drug in solid tumors and infected areas is increased, while the amount in normal tissue is decreased. This increases the therapeutic index of linked drugs. They appear to be a suitable medication delivery system because of their similarity to biological membranes and the fact that they are capable of taking in many different chemicals [101]. Many biomedical uses of liposomes have been developed during the past two decades, and some are now being tested in clinical studies [102]. As the liposome surface charges increase in density and complexity, so does the process through which they interact with cells. Both of these factors can be changed by altering the lipid composition. The liposomes might have a neutral, positive, or negative overall charge due to the inclusion of charged components in them. Increased liposome aggregation and decreased physical stability are both caused by neutral liposomes, which lack surface charge [103]. The extracellular space can discharge drugs from liposomes because neutral liposomes have little or no interaction with the cells that contain them [104]. 

In a study, Muthu et al. designed DTX-loaded liposomes coated with TPGS for brain tumor therapy. The solvent injection method was used to construct docetaxel or C6-loaded liposomes, which were then tested for various characterization parameters. PEG- or TPGS-coated liposomes had particle sizes between 126 to 191 nm. An electron microscope technique known as FETEM revealed that the liposomes were coated with TPGS [78]. In another study, Muthu et al. fabricated TPGS-coated theranostic liposomes co-loaded with docetaxel and quantum dots (QDs) for folate receptor-targeted breast cancer imaging and therapy. Cellular absorption and cytotoxicity of the medication and QD-loaded liposomes were evaluated in vitro using MCF-7 breast cancer cells with folate receptor overexpression. Liposomes with a non-targeting and a targeting function were discovered to have particle sizes of 202 and 210 nm, respectively. Using FETEM, QDs were found in the liposome’s hydrophobic membranes. CLSM revealed the qualitative uptake of multifunctional liposomes by MCF-7 cells. Non-targeting liposomes were found to be less effective than targeting liposomes and hence have considerable promise for improving cancer imaging and treatment [9]. A few years back, Sonali et al. fabricated transferrin receptor-targeted theranostic liposomes co-loaded with docetaxel and QDs for targeted brain cancer theranostics. Both non-targeted and targeted liposomes have particle diameters <200 nm. Using liposomes, over 71% of the drug encapsulation efficiency could be attained. In vivo data showed that adding transferrin receptor-targeted theranostic liposomes made it easier for docetaxel and QDs to get into the brain and pass through it [93].

Similarly, Muthu et al. formulated TPGS-coated theranostic liposomes of DTX and QDs. For the preparation of the formulations, solvent injection techniques were utilized. Cellular absorption and cytotoxicity of the medication and QDs loaded liposomes were evaluated in a folate receptor overexpressing breast cancer cell line, using an in vitro paradigm. Both the non-targeted and target liposomes had a mean particle size of 202 nm, although the latter was 210 nm. Figure 7I shows FETEM image of (A) single multi-purpose liposomes on the 100 nm scale, each with visible QDs in its bilayer membrane, and (B) innumerable liposomes on the 500 nm scale, which demonstrated the spherical nature of the liposomes. Moreover, the FETEM study validated the TPGS’s full distribution over the liposomes because no free TPGS was detected. CLSM was used to see how MCF-7 cells ingested multifunctional liposomes qualitatively. After 24 h in culture, MCF-7 cells had an IC_50_ of 9.54 ± 0.76, 1.56 ± 0.19 and 0.23 ± 0.05 mg/mL for the commercial Taxotere^®^, non-targeting liposomes, and targeting liposomes, respectively. The multifunctional, targeting liposomes performed better than the non-targeting liposomes and held much promise for advancing cancer diagnosis and treatment. Figure 7II(A) shows red intensity for QDs, and Figure 7II(B) for DAPI; Figure 7II(C) shows the merge channel for QDs and DAPI. The red fluorescence intensity and the amount of red staining in the cytoplasm of MCF-7 cells were both greater when treated with the targeting multifunctional liposomes (DTX-QDFA) than when treated with the non-targeting liposomes (DTX-QD) Figure 7II(C). In addition, it has been discovered that MCF-7 cells overexpress folate receptors in their cell membrane by a factor of more than 100, compared to normal cells [9].

### 7.3. TPGS-Based Polymeric Nanoparticles for Nanotheranostics

The development of cutting-edge technologies has resulted in the advancement of the novel nanotheranostic platform for the simultaneous diagnosis and treatment of various diseases, including cancer. Using biodegradable and biocompatible polymeric nanoparticulate systems, regulated drug delivery and drug targeting are viable possibilities. Solid colloidal polymeric NPs have a diameter of 1–1000 nm [105]. Because of their small particle size and extended blood circulation, they have been studied for their potential in medication delivery and therapeutic targeting. As adjuvants in vaccines and drugs, these macromolecular components can be utilized to dissolve, entrap, adsorb, or chemically bind the active ingredient. Drug delivery issues, such as those based on drug targeting and those requiring the delivery of undeliverable molecules like oligonucleotides or RNA interfering effectors, can be overcome using polymer nanotechnology [106]. NP preparation methods are a significant element of this problem. They enable the creation of polymeric NPs with the right characteristics to ensure the appropriate transport and targeting of drugs. Polymeric NPs may now be generated under controlled conditions due to improvements in polymer chemistry and polymer colloid physico-chemistry. NPs that target drugs can be made if polymers have well-defined structures and compositions and if it is possible to make NPs with very specific properties [107]. Polymeric NPs can release drugs in a number of ways, such as when the polymer swells, when drugs move through the polymer matrix, when the polymer erodes or breaks down, or when a combination of these things happens. Spawning of the polymer, generating holes, or drug diffusion from the polymer surface are all possible explanations for the initial burst release from the polymeric NPs. In addition, polymeric NPs have pH-dependent drug release as a result of their solubility [108]. In a study, Mehata et al. developed trastuzumab functionalized docetaxel-TPGS-g-chitosan NPs for the treatment of breast cancer. Conventional, non-targeted NPs had an entrapment effectiveness of 74–78%, while targeted NPs had an entrapment efficiency of 78–95%. They found that TPGS-g-chitosan NPs loaded with docetaxel were more effective at increasing cell uptake and killing cancer cells in vitro than standard formulations, such as Docel^®^. In a cytotoxic assay, it was discovered that the IC_50_ values of Docel^®^ were 43 and 223 times greater than those of non-targeted and targeted NPs, respectively. In addition, non-targeted and targeted NPs had half-lives that were 3.48 and 5.94 times longer than Docel^®^ after i.v administration. In histopathology tests, NPs that were either not targeted or were targeted did less damage to the organs such as lungs, liver, and kidneys, than Docel^®^ [66].

Further, Feng et al. formulated NPs using a modified solvent extraction/evaporation process that included TPGS or PVA as an emulsifier to encapsulate the paclitaxel-loaded PLGA. Fluorescent NP uptake was examined in rabbit carotid arteries and in vitro in coronary artery smooth muscle cells. NPs compositions were tested in vitro against Taxol^®^ to see if they had any antiproliferative effects. In terms of drug-encapsulation efficiency, cellular uptake, and cytotoxicity, the TPGS-emulsified NP formulations stand in stark contrast to the PVA-emulsified NP formulations. In a 24 h culture with coronary artery smooth muscle cells, paclitaxel, PVA-emulsified NPs, and TPGS-emulsified NPs have IC_50_ of 748, 708, and 474 ng/mL, respectively. In the future, NP-coated coronary stents made from TPGS-emulsified PLGA NPs could represent the third generation of cardiovascular stents, allowing for the effective and long-term delivery of antiproliferative drugs and other therapeutics [109].

Furthermore, Chauhan et al. developed a novel Toco-Photoxil plasmonic imaging technology for photothermal treatment that is biodegradable and biocompatible. For nanotheranostic applications, TPGS-modified gold was loaded with the Pgp inhibitor. Toco-Photoxil was designed to absorb at 750 nm in order to provide the best therapeutic results. According to in vivo research, the disintegration of Toco-Photoxil by NIR light irradiation causes heat to be generated, which kills cancer cells in the body. The system is quickly removed from the body as it disintegrates, reducing any potential toxicity problems that may arise as a result of material build-up. Reduced photothermal transduction capability was caused by folic acid (FA) and IR780 functionalization of Toco-Photoxil because the plasmon resonance was disrupted. A strong X-ray attenuation power allows Toco-Photoxil-imbedded phantoms to show comparable contrast to the iodine-based contrast agent Omnipaque at five times less concentration. Within 4 h, IR780-Toco-Photoxil began to accumulate in the HT1080-fluc2-turboFP tumor and achieved near saturation within 12 h. The brightest signal persisted for 48 h. The IR780-FA-Toco-Photoxil, which selectively kills tumor cells that lack a folate receptor, was also developed for comparison. When compared to IR780-Toco-Photoxil, it significantly reduced the amount of noise in the data; see Figure 8. At 24 h, IR780-FA-Toco-Photoxil was most enriched, and less dye leached to other body areas. Rapid accumulation of IR780 at the tumor site was observed in control mice injected with IR780 dye alone, peaking at 4 h, and then diffusing out from the tumor bed, leaving almost very little tumor-specific fluorescence at the 72 h time point Figure 8I(a,b). The same research was then conducted with the mouse breast cancer cell line 4T1 in an FR (+) orthotopic tumor model. NIRF imaging shows that in the 4T1 orthotopic tumor, a greater number of IR780-FA-Toco-Photoxil particles accumulated, in comparison to the FR (−) HT1080 xenograft model Figure 8I(a,c). The authors also performed in vivo photothermal therapy on BALB/c nude mice with the HT1080-fluc2-turboFP xenograft model to learn more about how well Toco-Photoxil works as a treatment and to find out what narrow and broad absorbance peaks mean for photothermal therapy in a physiological setting. On day 24, mice (n = 4) with advanced tumors were randomly assigned to receive either 750 nm tuned or 915 nm tuned Toco-Photoxil via intratumoral injection. Although a log-fold decrease in bioluminescence signal was seen in the 915 nm laser treatment group with an average radiance of 1.805 × 10^9^ ± 1.009 × 10^9^ p/sec/cm^2^/sr, this suggests less therapeutic efficacy was obtained in the same time period Figure 8II(a). In addition, bioluminescence imaging demonstrated that the tumor mass in group V did not completely retreat 15 days after photothermal therapy, while animals in group IV exhibited no symptoms of recurrence throughout the month-long follow-up period; see Figure 8II(b). Group IV animals that had their tumor loads reduced had a statistically significant increase in post-PTT Figure 8II(c). After the nanoshells were administered systemically, photothermal therapy was initiated through localized laser irradiation between 24 and 48 h, at which point the solid tumor mass was completely ablated due to the persistent and high EPR-mediated build-up of IR780-Toco-Photoxil. Additionally, TurboFP fluorescence and firefly luciferase bioluminescence reporter imaging protocols were performed to cross-validate tumor localization. The HT1080-fluc2-turboFP tumor’s TurboFP fluorescence signal in the IR780-Toco-Phtoxil treatment group was considerably lower than in the IR780 dye control group at 72 h post-PTT Figure 8III(a,b). Day 10 post-photothermal treatment in vivo non-invasive bioluminescence imaging was performed to confirm any sign of tumor cell resurrection at the location Figure 8III (c). There was only a slight recovery of the HT1080 tumor in the IR780-Toco-Photoxil group of rats compared to the IR780 dye control group Figure 8III(d). NP accumulation was shown to be substantially higher in the case of IR780-Toco-Photoxil (0.025% ID) than in the case of IR780-FA-Toco-Photoxil (0.005% ID), while the highest was recorded in bare Toco-Photoxil (2.3% ID) across all tumor models. Within 72 h of intravenous injection, the enrichment of Toco-Photoxil rose from 2.0% to 2.3%, but the increase in FA-Toco-Photoxil accumulation was negligible and barely reached 0.3%. Figure 8III(e,f). Toco-Photoxil passes all of the most important biocompatibility tests with excellent grades and has a low clearance inside the human body. Toco-Photoxil is a potent and superior PTT agent due to its high tumor-specific accumulation from systemic circulation, potent photothermal conversion, and very safe material properties in human physiology, all of which may one day lead to accelerated clinical testing [110].

Most recently, Vikas et al. formulated TPGS emulsified dual receptor (folate and EGFR) targeted chitosan NPs loaded with docetaxel for lung cancer therapy. The developed NPs were decorated with folic acid and cetuximab to target overexpressed folate and EGFR receptor, respectively, in lung cancer and the targeted delivery of docetaxel. Additionally, in vitro cytotoxicity study revealed that dual receptor targeted NPs have 34 times higher cytotoxic effects than control docetaxel. Further, in vivo pharmacokinetics study suggested that targeted NPs have improved bioavailability compared to docetaxel control. The improved cytotoxicity and bioavailability for the developed NPs can be attributed to the P-gp inhibition and enhanced solubilization effect of TPGS in the formulation [60].

### 7.4. TPGS-Based Quantum Dots for Nanotheranostics

Nanomaterials have a lot of potential for use in cancer treatment. One such example is QDs, with sizes between 2–100 nm and with uniquely adjustable optical and targeting capabilities [111]. The primary objective is to create miniaturized probes that are extremely selective, flexible, stable, and able to access deep within cells and organelles. Despite widespread excitement, advances in this area have been hampered by issues with reproducible production and biological obstacles. The majority of current research on QDs is directed toward the development of nanocrystals and their bioconjugation for tracking and imaging [112]. QDs have not been investigated for their potential as sensitizers in cancer treatment. Colloidal QDs are fluorescent dyes based on inorganic semiconductor NPs, in contrast to organic fluorophores. After being photo-excited, these particles form electron–hole pairs; when recombining, they emit light as fluorescence. Because of their nanoscale dimensions, quantum effects play a significant role, leading to fluorescence wavelengths that vary with particle size. The bluer the fluorescence, the smaller the particle size [113]. Thus, by manufacturing NPs of varying sizes, it is possible to obtain any color in the visible and infrared spectrums. When compared to organic fluorophores, QDs have various distinct photophysical characteristics. Their emission spectra are relatively narrow and symmetric; they do not exhibit any red-tailed emission, and their absorption spectra are continuous for wavelengths shorter than the wavelength of fluorescence emission. This allows for the spectrally clean excitation of a wide range of colors using a single excitation wavelength [114].

In a study, Su et al. developed a graphene quantum dot (GQD) nanoaircraft (SCNA) for the treatment of cancer. The nanoaircraft was built from GQDs with a size of less than 5 nm and a pH-sensitive polymer functionalized on their surface; it loses its stealth function in the mildly acidic tumor environment but retains stability at physiological pH. The SCNA is able to effectively penetrate deep tumor tissue owing to a size conversion triggered by NIR irradiation, which shrinks the 150 nm of the SCNA into the 5 nm of (DOX)/GQD. DOX/GQD can enter tumors and then spread to nearby cancer cells, killing them repeatedly. By combining SCNA with conventional treatment, xenograft tumors can be significantly diminished in just 18 days with minimal systemic side effects. Medications and energy are successfully delivered to the deep tumor using this intricate mechanism, showing its promise for use in a variety of tumor therapies. As shown in Figure 9I(a), tumor cells had taken up a considerable number of SCNAs (150 nm), but these SCNAs were mostly located on the multicellular tumor spheroids (MTSs’) periphery. Closer inspection revealed that the MTSs’ interiors were devoid of SCNAs except for a scant few. When compared, brightly fluorescent 5 nm GQDs were able to go deep within the MTSs Figure 9I(b). Many GQDs were visible even within the MTSs themselves. These results show that GQDs have higher cellular absorption efficiency and tumor penetrating abilities due to their shape and size. To verify that the GQDs were delivered intercellularly, a three-stage incubation was performed to examine the escape and was swallowed by cells of DOX/GQDs. To begin with, the DOX/GQDs were co-cultured with RG2 cells for 12 h. Figure 9I(c) demonstrates that CLSM can detect DOX fluorescence. After being exposed to NIR light for 5 min, the DOX/GQDs were released from the cells, centrifuged at 2000× *g* rpm for 5 min to separate them from the primary cells, and then reincubated with untreated, fresh cells. DOX/GQDs fluorescence via NIR was still detectable at the second step, indicating that the DOX/GQDs had been successfully liberated at the first stage, Figure 9I(c). DOX fluorescence was reduced in the second stage when NIR irradiation was absent from the DOX/GQD groups. Without NIR and DOX, just a few weak signals were obtainable for DOX/GQDs in the third stage. However, DOX intercellular distribution was only possible once NIR was applied to DOX/GQDs. For a given time period of cellular uptake, the GQDs treated with NIR had much higher fluorescence intensities than those without NIR, demonstrating the enhancement of intercellular transport by the photothermal effect Figure 9I(d). In addition, the GQD groups fared better than the non-GQD groups. Edge asperities and corner sites of the GQDs may boost intercellular transport rather than endocytosis due to their ability to speed up cellular absorption efficiency and intracellular permeation characteristics. Live/Dead assays were used to research the photothermal effects of the SCNAs. The RG2 cells were treated with SCNAs before and after being exposed to NIR light for 3 min at a power level of 2 W/cm^2^. Following treatments, cell viability was characterized by Live/Dead staining, in which dead cells were labeled red and live cells were stained green. The SCNA-only group had no signs of cell death before NIR irradiation, but after 3 min of NIR, the cells were visibly dead in certain areas due to the temperature increase, demonstrating the remarkable hyperthermia efficiency Figure 9I(e). The 100 µL of SCNA at 2 mg/mL was given intravenously to nude mice harboring the RG2 tumor to determine the efficacy of hierarchical targeting to in vivo cells. The in vivo IVIS was used to take pictures of the SCNAs’ biodistribution after 24 h had passed after injection. Based on IVIS data, it appears that the pH sensitivity improved accumulation by hierarchical targeting since SCNAs accumulated at a significantly higher rate than nonfunctional nanoaircrafts (NNAs) Figure 9II(a). In addition, IVIS spectral imaging was used to examine important organs and malignancies. Both SCNAs and NNAs showed robust fluorescence in the liver, lung, kidney, and tumor 48 h after injection Figure 9II(b). When compared to other key organs, IVIS revealed that the SCNAs’ cumulative efficiency in the tumor was 70%, while the NNAs’ efficiency was 40%. Reduced pH-sensitive accumulation in the reticuloendothelial system was also seen in the signals of metabolic organs such as the liver and spleen, Figure 9II(c). With SCNA-treated tumor-bearing mice, 2 W/cm^2^ of NIR irradiation was administered for 10 min, and the tumor was heated to 48 °C Figure 9II(d). The tumor temperature rose to 40.7 °C after DOX/NNA was applied. Alternatively, irradiated tumors in mice treated with saline or DOX showed no discernible increase in surface temperature. Since GQDs had the same photothermal conversion ability as SCNAs, the greater photothermal efficacy of the SCNAs may have been linked to greater tumor accumulation. Subsequently, there was no discernible difference between treated and untreated mice in terms of hematoxylin and eosin (H&E) staining of the primary organs, indicating a lack of influence on other organs, Figure 9II(e). However, photothermal therapy caused cracks, holes, and even fibrosis in the tumor, and photothermal therapy and chemotherapy worked better together, Figure 9II(f) [115].

### 7.5. TPGS-Based Miscellaneous Agents for Nanotheranostics

The organic and inorganic materials available for use in nanocarrier fabrication are extensive, including polymeric NPs, nanocapsules, micelles, liposomes, dendrimers, quantum dots, etc. Drugs are being encapsulated in these materials or made more soluble in order to facilitate in vivo drug delivery, and these materials are also being employed in imaging due to their distinct optical, magnetic, and electrical properties [116]. However, due to a lack of research into their potential dangers, NPs should be utilized with extreme care. Nanoparticles have potential medical applications, but their toxicity and safety need more investigation.

In a study, Singh et al. designed a more effective delivery system for the chemotherapy medicine DTX by coating or covalently conjugating carbon nanotubes (CNT) with TPGS and loading the NPs with docetaxel, a model drug for treating lung cancer in comparison with the marketed product (Docel^®^). After 24 h of treatment, the IC_50_ values for the TPGS conjugated CNT showed that it could be 80 times more effective than Docel^®^. Analyses using flow cytometry showed that the presence of TPGS-coupled CNT led to a substantial increase in the number of cancer cells found in the sub-G1 phase [93].

Furthermore, Zhu et al. developed DTX-loaded NPs via polydopamine (pD) and TPGS. Galactosamine was conjugated on the produced NPs (Gal-pD-TPGS-PLA/NPs) to increase DTX distribution by ligand-mediated endocytosis to liver cancer cells. When compared to TPGS-PLA/NPs, the size and morphology of pD-TPGS-PLA/NPs and Gal-pD-TPGS-PLA/NPs were drastically altered. Similar DTX release profiles were seen in in vitro tests between TPGS-PLA/NPs, pD-TPGS-PLA/NPs, and Gal-pD-TPGS-PLA/NPs. The cellular absorption efficiency of C6-loaded Gal-pD-TPGS-PLA/NPs was best in the liver cancer cell line HepG2, as determined by both CLSM and flow cytometry. Moreover, TPGS-PLA/NPs, pD-TPGS-PLA/NPs, and a commercially available DTX formulation, Gal-pD-TPGS-PLA/NPs loaded with DTX, suppressed the expansion of HepG2 cells (Taxotere^®^). The Gal-pD-TPGS-PLA/NPs have been shown to be tumor-targeted by in vivo biodistribution studies. Injecting DTX-loaded Gal-pD-TPGS-PLA/NPs into hepatoma-bearing nude mice was associated with the greatest reduction in tumor size, as determined by the in vivo antitumor effects investigation. Based on these findings, it appears that the Gal-pD-TPGS-PLA/NPs generated for this study particularly interacted with the hepatocellular carcinoma cells via ligand–receptor recognition and might be exploited as a potentially acceptable drug delivery system targeting liver malignancies. Galactosamine was conjugated to the produced NPs (Gal-pD-TPGS-PLA/NPs) to increase DTX distribution by ligand-mediated endocytosis to liver cancer cells. When compared to TPGS-PLA/NPs, the size and morphology of pD-TPGS-PLA/NPs and Gal-pD-TPGS-PLA/NPs were drastically altered. Similar DTX release profiles were seen in vitro tests between TPGS-PLA/NPs, pD-TPGS-PLA/NPs, and Gal-pD-TPGS-PLA/NPs. The cellular absorption efficiency of C6-loaded Gal-pD-TPGS-PLA/NPs was best in the liver cancer cell line HepG2, as determined by both CLSM and flow cytometry. Moreover, TPGS-PLA/NPs, pD-TPGS-PLA/NPs, and a commercially available DTX formulation, Gal-pD-TPGS-PLA/NPs loaded with DTX, suppressed the expansion of HepG2 cells. The Gal-pD-TPGS-PLA/NPs are tumor-targeted by in vivo biodistribution studies. Injecting DTX-loaded Gal-pD-TPGS-PLA/NPs into hepatoma-bearing nude mice was associated with the greatest reduction in tumor size, as determined by the in vivo antitumor effects investigation. Based on these findings, it appears that the Gal-pD-TPGS-PLA/NPs generated for this study particularly interacted with the hepatocellular carcinoma cells via ligand–receptor recognition and might be exploited as a potentially acceptable drug delivery system targeting liver malignancies. As seen in Figure 10I,II, a modest percentage of the IR-780-loaded TPGS-PLA/NPs and Gal-pD-TPGS-PLA/NPs were found in the lung and tumor tissue 0.5 h after injection, while the fluorescence signals of free IR-780 demonstrated whole body distribution. Strong signals for free IR-780, IR-780-loaded TPGS-PLA/NPs, and Gal-pD-TPGS-PLA/NPs were found in the tumor tissue 6 h after injection. Gal-pD-TPGS-PLA/NPs that were loaded with IR-780 had a greater tumor signal than TPGS-PLA/NPs that were loaded with IR-780 or free IR-780. For the latter groups, the signal intensity in the liver and lungs was still quite high. After 24 h post-injection, all three groups of tumor tissue showed substantially stronger signal intensity than their lung and liver. Since ASGPR is frequently overexpressed by hepatocytes and hepatoma cells, it may be inferred that Gal-pD-TPGS-PLA/NPs have the ability to target the liver and hepatoma. Gal-pD-TPGS-PLA/NPs demonstrated the most impressive hepatoma targeting, which was quite exciting. Gal-pD-TPGS-PLA/NPs started to accumulate in the tumor at 0.5 h after treatment, and additional examination of the major tissues at 24 h revealed that the fluorescent signal in the tumor was unmistakably stronger than that in the liver, Figure 10III,IV. Gal-pD-TPGS-PLA/NPs are thought to be a specific ligand of ASGPR, and this ligand/receptor interaction allows these NPs to selectively concentrate in tumors [117].

## 8. Therapies Involving TPGS-Based Materials

### 8.1. Radiotherapy

In the past decade, the idea of nano-radio sensitization has exploded in popularity. One way of radiosensitizing tumors involves the use of nanomaterials with a high atomic number (high Z) to increase the amount of radiation deposited inside cells. Many novel nanomaterial-mediated radiosensitive approaches have been reported in rapid succession in recent years [118]. These methods may provide a fresh opportunity for improving radiotherapy (RT) outcomes while decreasing their potential risks. Nanomaterials with intrinsic radiosensitivity have been shown in a number of studies to increase the efficiency with which ionizing radiation is deposited, as well as to catalyze the generation of ROS, to regulate the tumor microenvironment (TME), and to manipulate the cellular circuitry and signalling pathways of cancer cells [119]. In addition to tumor radio sensitization strategies that rely on the inherent radiosensitive properties of nano radiosensitizers, strategies that use nanoparticles as nanocarriers to deliver a wide range of therapeutic drugs into tumors, such as nanomaterial-mediated synergistic RT, are promising developments [120]. The following conditions must be met by a perfect distribution method: selective accumulation and therapeutic action in tumor rather than normal tissue to lessen unwanted side effects; vascular delivery of therapeutic agents without drug leakage [121]. As a result of their high drug-loading capacities, excellent biocompatibility, and EPR effects, nanoparticles are a promising new drug delivery technology. Nanoparticles can be used to deliver a variety of radiosensitive therapeutic agents (including some chemotherapeutic drugs, gene materials, and photothermal agents) directly into tumors, where they can work together to increase the effectiveness of RT by making cells more vulnerable to radiation-induced damage by a variety of mechanisms [122]. Synergistic chemotherapy with RT is possible, for instance, when the chemotherapy medicines used have the ability to stop the cell cycle at the radiosensitive G2/M phase. Synergistic gene therapy with RT is largely due to gene materials that suppress the production of radioresistance-linked proteins. In addition, photothermal agents can achieve synergistic PPT with RT because they generate heat, which speeds up intratumoral blood flow and improves tumor oxygenation, making the cells more sensitive to RT [123].

In a study, Du et al. developed chalcogenide nanocrystals with TPGS (TPGS-Cu3BiS3). The TPGS-Cu3BiS3 NCs take advantage of the second NIR window’s strong absorbance of both X-rays and NIR light to deposit a higher radiation dose and help enhance tumor oxygenation. Copper ions on the surface of nanocrystals can catalyze Fenton-like and Haber–Weiss processes, increasing the number of oxygen radicals and so enhancing their ability to destroy cancer cells. Not only can TPGS-Cu3BiS3 NCs be used therapeutically through X-ray CT imaging and multispectral optoacoustic tomography imaging, but they also have prospects as a nanotheranostic that can provide notable therapeutic efficacy for depth cancer cells in imaging-guided synergistic effects augmented radiotherapy. By focusing radiation energy within tumor cells, high-Z elements can improve the efficacy of RT. In this work, they rationally construct Cu3BiS3 NCs that can perform deep PTT in the second NIR window and rapid thermal annealing. The first step was the hot-injection synthesis of OM-Cu3BiS3 NCs Figure 11I. Both the first and second NIR window PA signals are recorded with a varying number of nanocrystals, as shown in Figure 11II, and linear relationships between concentration and PA signals are observed. Intriguingly, nanocrystals can be used as a more effective PA probe in the second NIR window because their signals should not be disrupted by haemoglobin and oxyhaemoglobin. In the MSOT image taken prior to injection, the contrast of tumor blood shows up as faint PA signals in the tumor location, Figure 11II(c,d). After injection, tumor contrast increases dramatically, demonstrating the NCs’ superior contrast capabilities in MSOT imaging. By comparison, nanocrystals show strong CT contrast performance because bismuth has a greater absorption coefficient than iodine. Figure 11III demonstrates that as nanocrystal concentration is increased, HU values increase linearly. When compared to iopromide, the slope of ~9.25 is significantly steeper. More importantly, the in vivo result demonstrates that significant CT contrast signals in tumors are observed 3 h post-injection of the dispersion of nanocrystals, indicating that nanocrystals can be used as an efficient CT contrast agent Figure 11III(c,d). Based on these findings, nanocrystals may be used to create a multimodal contrast agent that combines CT and PA imaging techniques, thereby offering the possibility of accurate guidance during therapy [124].

### 8.2. Positron Emission Tomography

Tumors, cardiovascular disease, and neurological diseases can all be detected with remarkable accuracy using a PET scan. The metabolic or biochemical function of the tissues and organs can be seen on a PET scan, an imaging test. The PET scan employs a radioactive substance (tracer) to display metabolic activity, both typical and abnormal [125]. PET scans are often able to detect the abnormal metabolism of the tracer in diseases before other imaging procedures, such as CT scans and MRIs, can detect the disease itself [126]. Tracers are often injected into a vein in the hand or arm. It is common for the disease to be localized after the tracer has collected in parts of the body with elevated metabolic or biochemical activity. PET images are commonly used in conjunction with other imaging modalities to create what are known as PET-CT or PET-MRI scans [127].

Wang et al. developed nanorods based on calcium phosphate and modified with TPGS and S-thaumatin peptide for tigecycline administration and the treatment of pneumonia caused by tigecycline-resistant *Klebsiella pneumoniae*. Due to the targeting ability of S-thaumatin and the inhibitory effect on efflux pumps exerted by TPGS, tigecycline accumulation in bacteria is increased following incubation with the produced nanorods. S-thaumatin and tigecycline have a synergistic antibacterial capability, which boosts the nanorod’s antibacterial activity and allows them to overcome *Klebsiella pneumoniae’s* tigecycline resistance. Mice with pneumonia have a far better chance of survival after receiving intravenous injections of nanorods due to the dramatic reduction in white blood cell and neutrophil counts, the reduction in bacterial colonies, and the amelioration of neutrophil infiltration events. These results have therapeutic potential for treating illnesses caused by bacteria that are resistant to multiple classes of antibiotics. Utilizing a TPGS-based and Ts peptide-modified Cap nano-drug delivery system, they built Ts-TPGS/Cap/TIG (TTCT) for TIG (tigecycline) loading and application in pneumonia treatment. TIG can be delivered specifically to bacteria after being modified into a Ts peptide and then incubated with the bacterium. The nano-drug delivery system’s small particle size aids in drug penetration through cell walls, and TPGS has an inhibitory effect on efflux pumps. Therefore, the antibacterial action of TTCT can be improved by increasing TIG accumulation inside bacteria. The antibacterial activity of TTCT is further improved by the synergistic effect of Ts and TIG Figure 12I. Thus, medication resistance can be overcome, and effective treatment for TRKP-induced pneumonia can be achieved. Figure 12II demonstrates that 18FDG uptake was much higher in the lungs of those infected with KPN or TRKP. Reduced standardized uptake values for 18FDG show that the levels of 18FDG uptake dropped after infected mice were given TIG, TTCT, or Ts-TPGS/Cap [128]. 

### 8.3. Photothermal Therapy

Near-infrared (NIR) light is used in photothermal therapy (PTT) to cause tumor tissue to heat up and kill cancer cells [129]. NIR absorbents are utilized to improve thermal efficiency. DDSs may pave the way for PTT, which is both very effective and safe. They do this by making it easier for heat to be created in tumor tissue while reducing the damage that photothermal damage does to normal tissue nearby [130]. DDS-type NIR absorbents are used in PTT, in which a tumor is exposed to a beam of NIR light at a predetermined time after the absorbent has been injected. Before exposing a sample to NIR light, irradiation parameters, including fluence rate and irradiation duration in PTT, are often established. However, it is challenging to exert a maximum therapeutic impact with fixed irradiation parameters before PTT because reliable tumor-related information is nearly impossible to collect [131]. Accordingly, the antitumor efficacy of PTT has been improved by employing irradiation parameters that raise the intensity rate or prolong the irradiation time. In order to avoid a suboptimal outcome caused by insufficient irradiation, a technique of producing an excessive dose of irradiation has been created due to the challenges in implementing NIR irradiation with appropriate parameters depending on each particular tumor. The normal surrounding tissues are also easily damaged due to the high temperatures employed to treat the tumor [132]. Collateral thermal damage, including burns, inflammation, and edema, to tissue surrounding the tumor, has been recorded while using an NIR laser at a high fluence rate. Since most proteins are denatured at 60 °C, heating tissue to this or a higher temperature will always result in coagulative necrosis [133].

In one study, Rodrigues et al. prepared mesoporous silica shell-gold (Au-MSS) nanoparticles compiled carrier with PTT to improve their therapeutic potential. Electrostatic and chemical linking methods for functionalizing Au-MSS nanorods with TPGS and PEI were investigated in this study. To do this, TPGS and PEI were chemically linked to one another or changed with 3-(triethoxysilyl)propyl isocyanate. The manufactured Au-MSS nanorods are consistently shaped and have clearly delineated gold cores and silica shells. The synthesis procedure also determined the surface charge of the particles. In contrast to the positively charged nanoparticles produced by the formulations created through chemical linkage (Au-MSS/TPGS/PEI), the particles modified through electrostatic interactions (Au-MSS/TPGS-PEI) were slightly negative. FTIR and thermogravimetric analyses verified that the polymers had been successfully incorporated. The PTT capability of the particles was also unaffected by the Au-MSS functionalization. However, the drug encapsulation efficiency was lower in the AuMSS/TPGS/PEI nanorods. Au-MSS was found to be cytocompatible up to 200 µg/mL in vitro experiments; however, the biocompatibility of positively charged formulations was observed only up to 100 and 125 µg/mL. In sum, the obtained data verify the effective modification of Au-MSS nanorods with TPGS and PEI, and their usefulness as PTT and drug delivery agents. First, the UV-Vis-NIR absorption spectra of Au-MSS derivatives were acquired to evaluate their viability as a PTT application Figure 13II. Absorption spectra of Au-MSS nanorods show transverse and longitudinal resonances at 515 nm and 750 nm (i.e., NIR area), respectively. As an added bonus, the nanorods’ absorption capacity was not significantly altered after functionalization with either TPGS-PEI, TPGS-TESPIC, or PEI-TESPIC. The use of Au-MSS compounds in PTT is bolstered by their high absorption between 700 and 900 nm. Since biological components have a limited absorption of this radiation, we can also anticipate fewer off-target interactions. After establishing that Au-MSS derivatives could be excited by NIR light, they looked into their ability to transform that light into heat by monitoring the rise and fall in temperature that was generated by irradiation with a NIR laser Figure 13I. Figure 13III shows that after being exposed to a NIR laser for up to 10 min, all Au-MSS derivatives showed an increase in temperature. All of the formulations were also able to generate a temperature variation of around 40 °C; therefore, the addition of polymers had no discernible effect on the results. An increase in temperature can kill cancer cells by denaturing their proteins, rupturing their membranes, and disrupting their metabolic processes. Additionally, it is noteworthy that the performance of the Au-MSS/TPGS-PEI, AuMSS/TPGS/PEI (1:1), and AuMSS/TPGS/PEI (3:1) nanorods was not impacted even when the experiment was carried out in complex media Figure 13III,IV. These results are encouraging because particle aggregation can alter the nanorods’ absorption spectra and, in turn, their PTT capability. CLSM was used to examine the cellular absorption of nanoparticles after the cytocompatibility of Au-MSS formulations had been determined. A major challenge for cancer medicine delivery systems is increasing nanoparticle absorption by cells. The nanoparticles in this investigation were prepared by labeling Au-MSS nanorods with FTIC. All Au-MSS formulations show internalization in Figure 13V. These results are consistent with those of prior studies that indicated the nanorods’ ability to transpose the cell membrane, sometimes with greater effectiveness than their spherical counterparts. The CLSM pictures shown in Figure 13V further show that the absorption of the nanorods is enhanced after they have been functionalized with TPGS and PEI. The neutral surface charge of the nanorods may explain why cells treated with Au-MSS/TPGS-PEI and Au-MSS/TPGS/PEI (3:1) show identical staining by the nanorods. Contrary to expectations, the highly positively charged Au-MSS/TPGS/PEI (1:1) formulation appears to have lower internalization on HeLa cells than the other coated nanorods. The greater PEI content, which has been linked to cell membrane rupture and worse cytocompatibility, may account for such a finding. Collectively, these results suggest that cellular absorption of functionalized Au-MSS nanorods is not affected by the manufacturing process employed (electrostatic interaction vs. chemical linkage). HeLa cells were also able to internalize the Au-MSS/TPGS-PEI and Au-MSS/TPGS/PEI formulations, which bodes well for the drug’s long-term stability and therapeutic potential [134].

## 9. Randomized/Clinical Prospects of TPGS, and Its Patents

In a study, Caruso et al. examined the clinical result of corneal cross-linking treated with either the standard Dresden (sCXL) or the accelerated custom-fast (aCFXL) UV light. Riboflavin-TPGS is used in an irradiation treatment for advanced keratoconus. Before treatment and at the 2-year follow-up, 54 eyes from 41 patients were assigned to one of two CXL procedures. The sCXL group underwent CXL that included a 30 min pre-soaking period and 30 min UVA irradiation at a dose of 3 mW/cm^2^. The aCFXL group underwent CXL with a 10 min pre-soaking followed by UVA irradiation at 1.8 ± 0.9 mW/cm^2^ for 10 min and 1.5 min. In both sets of subjects, a riboflavin-TPGS solution was administered. Baseline and follow-up 24-month evaluations included testing for uncorrected and corrected distance visual acuity, pachymetry, Scheimpflug tomography, and corneal hysteresis. No statistically significant variations in results were found between the groups, and both demonstrated statistically significant improvements in corrected distance visual acuity as well as keratometric and corneal hysteresis compared to pre-treatment levels. After sCXL and aCFXL, refractive, topographic, and biomechanical parameters all improved, showing the riboflavin-TPGS solution as a viable alternative as a permeation enhancer in CXL treatments. Riboflavin may provide further photo-protection by neutralizing ROS and UVA radiation produced by photo-induced activities, allowing it to penetrate deeper into the stroma [135].

In another study, Ozates et al. compared the effects of topical coenzyme Q10 (CQ10) and vitamin E on oxidative stress markers, including superoxide dismutase and malondialdehyde, in eyes with pseudo-exfoliative glaucoma (Coqun drop). Sixty-four eyes from sixty-four patients participated in this prospective, randomized clinical research. Surgical procedures, including phacoemulsification and intraocular lens implantation, were performed on all patients. Prior to beginning cataract removal, the surgeon collected a sample of the aqueous humor from the eye’s anterior chamber. Patients with pseudo-exfoliative glaucoma were divided into two groups: those who used Coqun (100 mg CQ10, 500 mg TPGS) topically twice daily for one month prior to surgery and those who did not take Coqun. The pseudo-exfoliation syndrome group consisted of individuals who had symptoms consistent with the condition. Results from both groups’ aqueous humor levels of superoxide dismutase and malondialdehyde were the primary indicators. The pseudo-exfoliative glaucoma group had significantly higher mean aqueous humor superoxide dismutase levels than the pseudo-exfoliative glaucoma+Coqun and pseudo-exfoliation syndrome groups. The results indicate that patients with pseudo-exfoliation syndrome have a lower amount of superoxide dismutase in their aqueous humor than those with pseudo-exfoliative glaucoma. Patients with pseudo-exfoliative glaucoma who were treated with topical Coqun had significantly lower superoxide dismutase levels than those who were not treated with Coqun. During the subsequent month of monitoring, there was no discernible rise or fall in malondialdehyde levels [136]. The description of various clinical trials of the TPGS polymer is illustrated in Table 3. Herein are offered compositions that include TPGS and/or its analogs. Both the dimer and monomer forms of the vitamin E derivative can be found in water-soluble vitamin E mixtures. Several lists of patents of TPGS-based materials are provided in Table 3.

## 10. Regulatory Status of TPGS and Marketed Formulations

TPGS was approved by USFDA in 1998 as an inactive ingredient under the name “Tocophersolan” and monograph has also been included in the USP/NF. Vit. E TPGS is not regarded as an active drug, but ISOCHEM owns one active Type II Drug Master File that is registered with the FDA. Eastman, TPGS’s inventor, has a self-proclaimed “generally recognized as safe” (GRAS) status, which has never been challenged by the USFDA. TPGS has been authorized for numerous specialized product categories since 1999. Since then, four drugs with TPGS in their preparations have been approved for sale in the United States and Europe. TPGS is also the main component in Vedrop, which is a vitamin E supplement [33].

The grand breakthrough was achieved when two pharmaceutical formulations containing “amprenavir”, an HIV protease inhibitor, were approved for marketing in 1999. It was observed that TPGS produces nanomicelles of amprenavir and enhances its water solubility from 36 µg/mL to 720 µg/mL. The above results helped in the development of two commercial products (soft gelatin capsule and nano solution) containing amprenavir by GlaxoSmithKline [37,146]. This is yet another method for making a weakly water-soluble medication more soluble, including Paclitaxel solubility improvement from 1.34 µg/mL to 50 µg/mL in water at 37 °C by the usage of 5 mg/mL of TPGS [147]. These two case studies show how TPGS uses its surfactant characteristics to enhance the solubilization of poorly soluble drugs in marketed products. The usage of TPGS, on the other hand, is not restricted to simply solubilization improvement. The preceding interpretations provide a broad overview of the various applications for TPGS-based revolutionary drug delivery solutions [148].

## 11. Conclusions and Future Perspectives

TPGS has firmly demonstrated itself as a viable choice for drug delivery advancement because of its nonionic surfactant qualities, solubilizer, emulsifier, and stabilizer properties, and even cellular penetration and uptake-enhancing effects. In the early 2000s, the first surge of pharmaceutical formulations premised on TPGS solubility improvement emerged. Parallel to this, the popularity of TPGS is reflected in the constant growth in patent filing and scholarly published articles. Recently, TPGS has been utilized in a huge number of Phase I, II, and III clinical trial studies. Additionally, numerous TPGS-based nanotheranostics reported in the literature are in the preclinical stage. In the near future, the second wave of drug products will be approved, in which TPGS will play an important role in improving drug efficacy and diagnostic properties. With its emergence as a water-soluble type of vitamin E, it was noted that TPGS has wider applicability not only to the pharmaceutical industry but also to the nutrition and food industries. Commercially, TPGS is available in the global market due to its safety and nontoxic properties, and its ease of handling during the processing and preparation of the drug delivery system. TPGS has powerful adsorption characteristics that reduce the potential for particle development and aggregation in multi-particulate formulations. Chemically modifying the PEG terminal of the TPGS with targeting moieties allows for more control and site-specific localization of the payload. TPGS-based nanomedicines functionalized with targeting moiety have the ability to preferentially relocate to the tumor location and produce lethal effects on tumor cells. Numerous studies have depicted that TPGS has been found to increase ROS generation and lower the concentration of the efflux pump proteins (P-gp) inside the tumor cells, enhancing the anticancer effect of therapeutic agents. Moreover, simultaneous activation of multiple death processes in tumor foci has been shown to improve response to treatment in cell lines and animal studies. Thus, TPGS has proven its importance in the advancement of drug delivery systems ranging from widely used treatments to highly developed nanotheranostics for treating a wide variety of diseases such as cancer, tuberculosis, inflammatory bowel disease, ocular diseases, neurodegenerative and inflammatory diseases, etc., due to its novel, amphiphilic, biocompatible, biodegradable, nontoxic, and nonimmunogenic nature. 

## Figures and Tables

**Figure 1 pharmaceutics-15-00722-f001:**
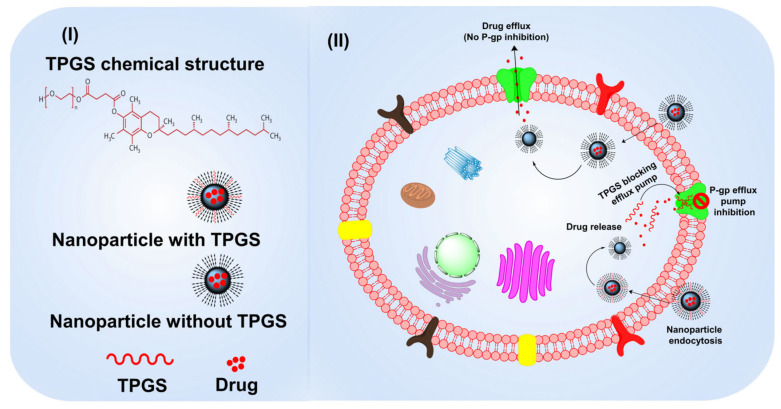
The medicinal applications of TPGS synthesized from vitamin E; (**I**) TPGS structure; and drug-loaded nanoparticles with or without TPGS; (**II**) Detailed explanation of the cellular mechanism by which TPGS inhibits P-gp.

**Figure 2 pharmaceutics-15-00722-f002:**
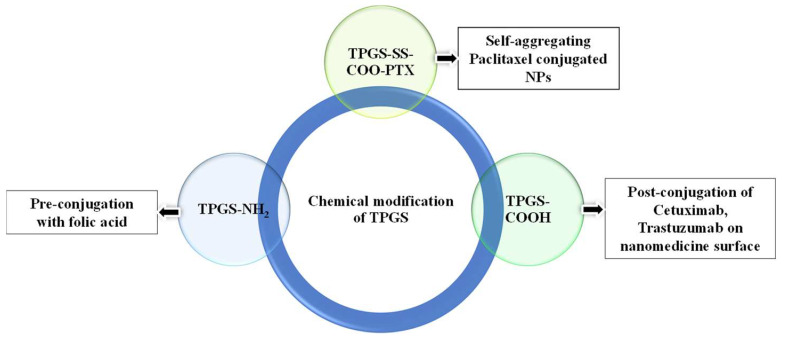
Various examples of chemically modified TPGS in cancer drug-delivery systems.

**Figure 3 pharmaceutics-15-00722-f003:**
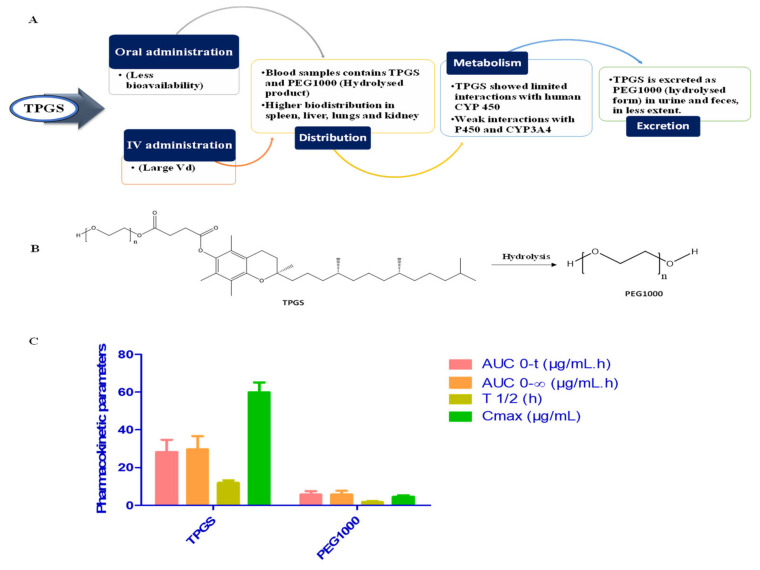
(**A**) Various pharmacokinetic aspects of the TPGS, (**B**) Hydrolysis of the TPGS converting into PEG1000. (**C**) Various pharmacokinetic parameters of the TPGS and its metabolites after intravenous administration of TPGS in rats at the dose of 5 mg/Kg.

**Figure 5 pharmaceutics-15-00722-f005:**
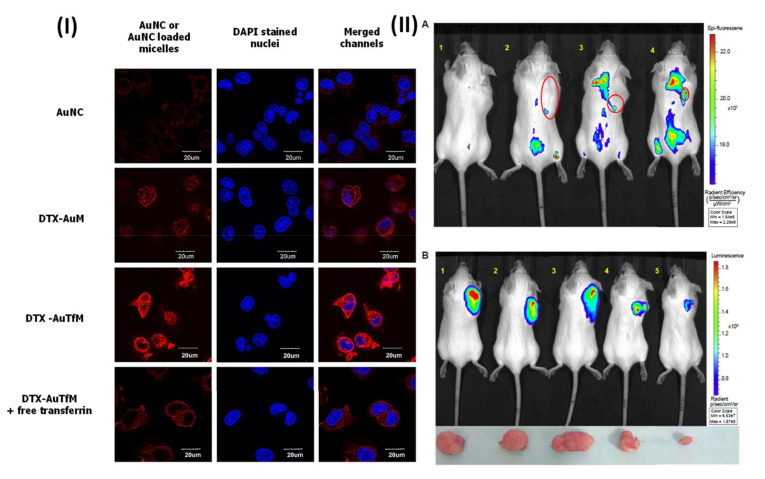
(**I**) CLSM photograph in breast cancer cells; (**II**) (**A**) Mice with tumors imaged and treated with fluorescent targeting fluoroscopy; (**II**) (**B**) Bioluminescent photos of tumor regression in live mice. Reproduced with the permission from Ref. [86], Figures 5 and 8 (Elsevier 2015).

**Figure 6 pharmaceutics-15-00722-f006:**
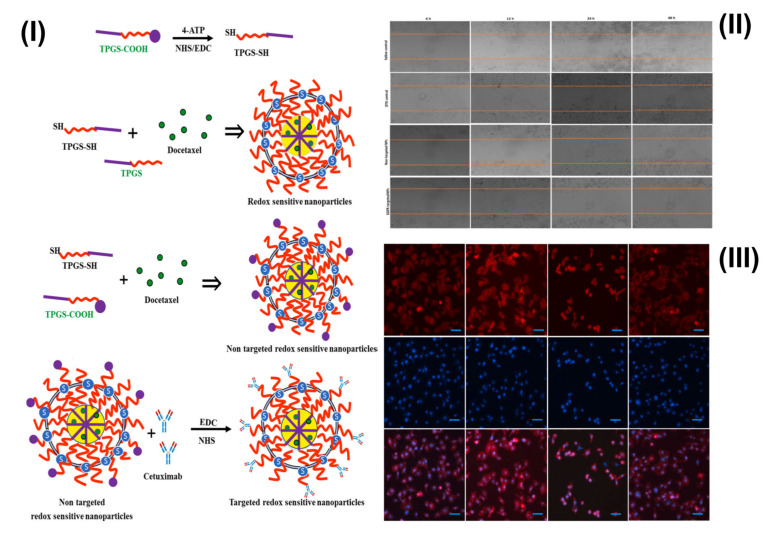
(**I**) Method for making redox-sensitive NPs that can be used in specific applications are described; (**II**) Wound healing assay (in vitro); (**III**) apoptosis studies (in vitro). Reproduced with permission from Ref. [61], Figures 1 and 6 (Elsevier 2021).

**Figure 7 pharmaceutics-15-00722-f007:**
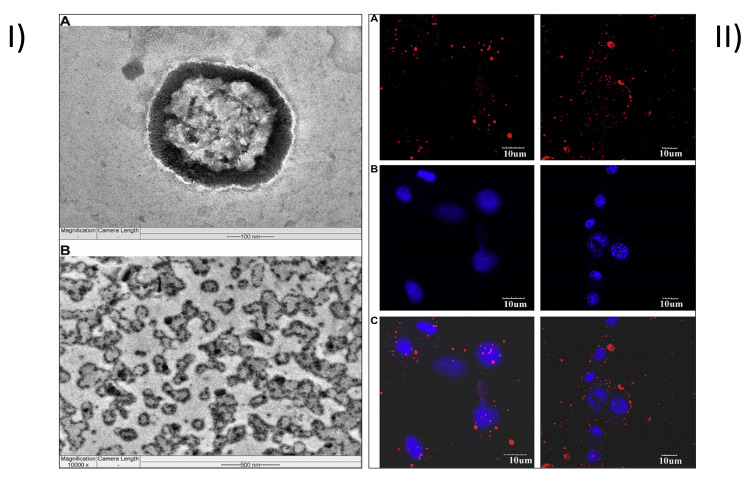
(**I**) FETEM images of QDs; (**A**) liposomes filled with separate QDs and coated with TPGS, (**B**) multiple QDs loaded TPGS coated liposomes (**II**) CLSM photograph of MCF-7 cells (**A**) The red fluorescence of liposomes in the cytoplasm is depicted by quantum dots; (**B**) DAPI channels; (**C**) Merged channels of QDs and DAPI. Reproduced with permission from Ref. [9], Figures 3 and 6. (Elsevier^©^ 2012).

**Figure 8 pharmaceutics-15-00722-f008:**
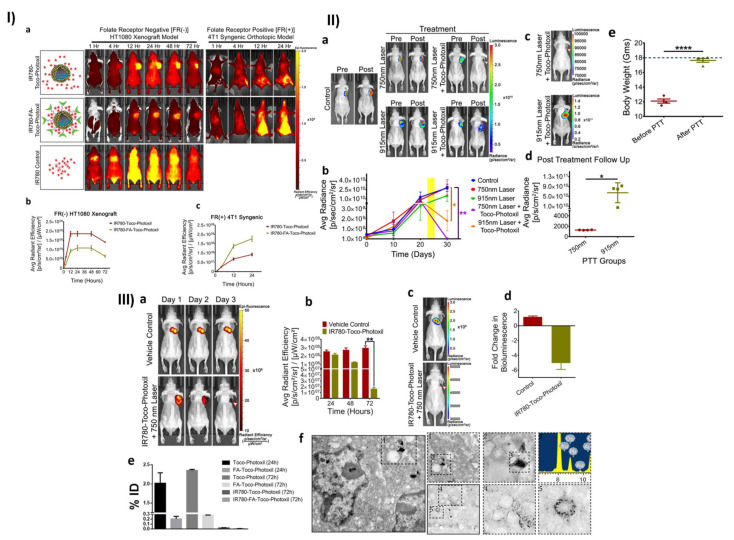
(**I**) (**a**) Near-Infrared Fluorescence (NIRF) imaging in tumor-bearing mice; (**b**,**c**) Quantitative assessment of fluorescence. (**II**) (**a**) In vivo bioluminescence imaging of tumor-bearing mice; (**b**) Bioluminescence signal quantification as an indicator of therapeutic success; (**c**) Bioluminescent pictures of mice after therapy; (**d**) Bioluminescence signal quantification; (**e**) Body weight before and after photothermal treatment is shown on a graph. (dotted line represents the average body weight of a non-tumor bearing BALB/c NUDE mice) (* *p* < 0.05, ** indicate *p* < 0.001, **** indicates *p* < 0.0001) (**III**) (**a**) Expression of the quality of mouse TurboFP fluorescence images; (**b**) Quantitative evaluation of TurboFP fluorescent protein light output variations; (**c**) Photographs of mice showing bioluminescence after treatment; (**d**) Quantifying the relative increase or decrease in bioluminescence between vehicle-treated animals versus mice treated with a mixture; (**e**) Cancer prevalence as a percentage of total tumor uptake; (**f**) The 4T1 tumor segment was imaged using fluorescent electron microscopy and transmission electron microscopy (FEG-TEM). Reproduced with permission from Ref. [110], Figures 6–8. (Nature^©^ 2018).

**Figure 9 pharmaceutics-15-00722-f009:**
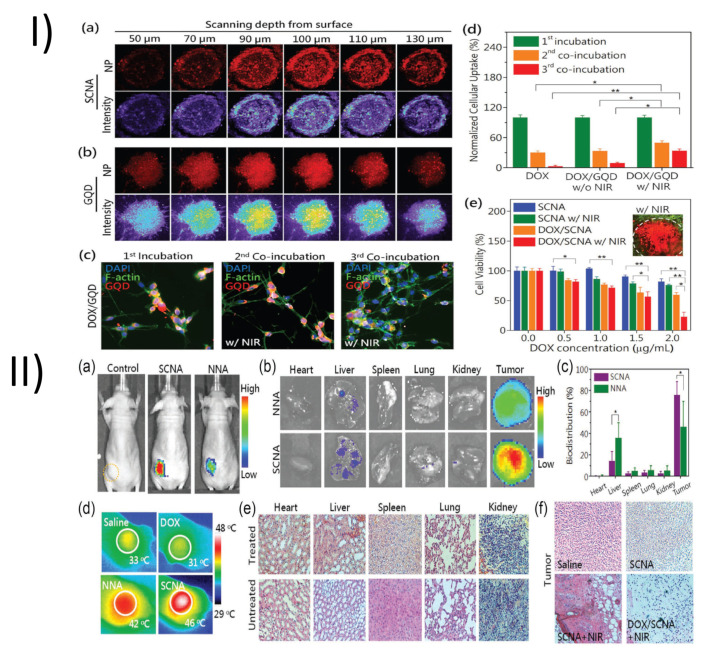
(**I**) (**a**,**b**) In an ex vivo penetration assay, photographs of fluorescence-labeled SCNAs and GQDs in RG2 multicellular tumor spheroids were used; (**c**) DOX/GQDs doubly coincubated with NIR exhibited a cellular sequence of intercellular delivery; (**d**) Flow cytometry standardizes the amounts of intercellular delivery by dividing by the number of cells and how well they take up the substance after the first incubation; (**e**) The cytotoxicity of SCNA and DOX/SCNA treatments for RG2 cells, with and without 5 min of NIR irradiation, were compared. (**II**) (**a**) IVIS image of tumor-bearing mice; (**b**) Mice tumors and primary organs involved in their clearance, as shown by fluorescence; (**c**) Fluorescence intensity of major organs and tumors; (**d**) Naked mice with RG2 tumors, as seen by thermography; (**e**) Histological examination of major organs in untreated and treated mice; (**f**) H&E tumor staining after photothermal therapy combined with chemotherapy for SCNA showed pore damage and fibrosis. (* Indicate *p* < 0.05 and ** indicate *p* < 0.001). Reproduced with permission from Ref. [115], Figures 5 and 6. (Wiley^©^ 2017).

**Figure 10 pharmaceutics-15-00722-f010:**
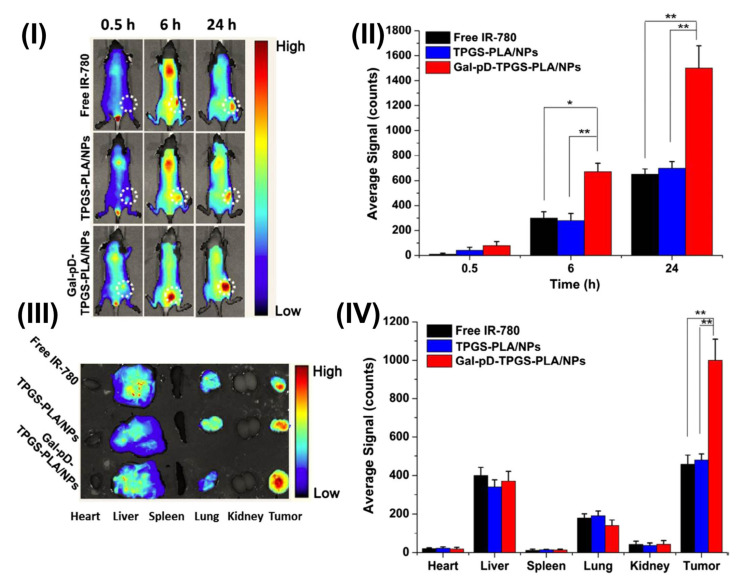
Studies of biodistribution and in vivo imaging were performed on nude mice; (**I**) NIR fluorescence time-lapse pictures of naked mice; (**II**) Tumor NIR fluorescence intensity was measured at the specified time points; (**III**) Major organs and malignancies in NIR fluorescence pictures after injection; (**IV**) Semiquantitative 24 h biodistribution in nude mice. (* Indicate *p* < 0.05 and ** indicate *p* < 0.001) Reproduced with the permission from Ref. [117], Figure 7 (Elsevier^©^ 2015).

**Figure 11 pharmaceutics-15-00722-f011:**
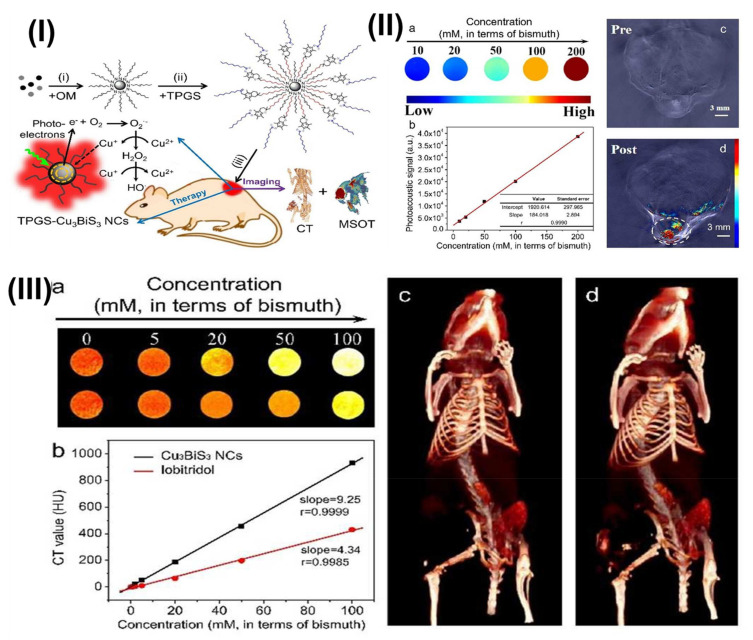
(**I**) Radiation therapy could be improved with the use of TPGS-functionalized Cu3BiS3 NCs, as shown in this schematic; (**II**) (**a**) Aqueous dispersion photos of TPGS-Cu3BiS3 NCs at various concentrations captured using a PA phantom; (**b**) Diagrams depicting the PA signal’s relationship to TPGS-Cu3BiS3 NC concentration; (**c**,**d**) In vivo PAT imaging of a tumor-bearing mouse before (**c**) and after (**d**) Injection with TPGS-Cu3BiS3 NCs; (**III**) (**a**) Cu3BiS3 NCs (upper) and iopromide (lower) CT phantom images at varying doses; (**b**) Diagram showing the relationship between TPGS-Cu3BiS3 NCs, iopromide HU values, and sample concentrations; (**c**,**d**) CT scans of mice taken prior to the injection of TPGS-Cu3BiS3 NCs; (**c**) and 3 h after intravenous injection of TPGS-Cu3BiS3 NCs (**d**). Reproduced with permission from Ref. [124], Scheme 1, Figures 6 and 7. (Royal Society of Chemistry^©^ 2017).

**Figure 12 pharmaceutics-15-00722-f012:**
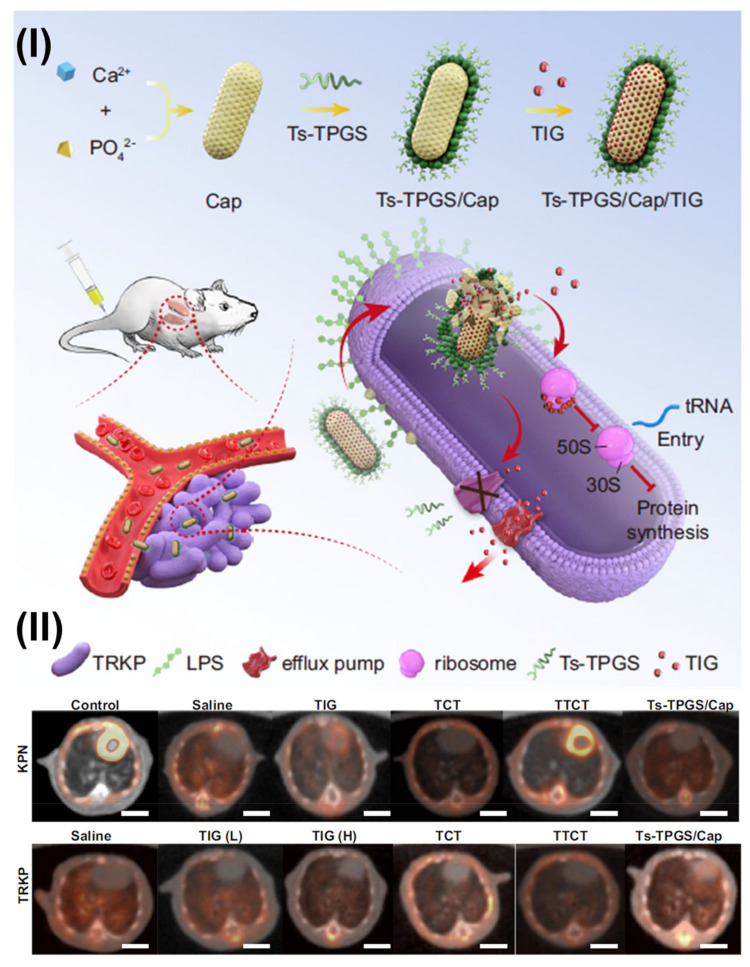
(**I**) Schematics show how the nanorods are put together and how the TRKP’s resistance to tigecycline can be broken; (**II**) A comparison of PET-CT scans of the lungs taken before and after dosage. Reproduced with permission from the Ref. [128], Figures 1 and 8. (Nature^©^ 2022).

**Figure 13 pharmaceutics-15-00722-f013:**
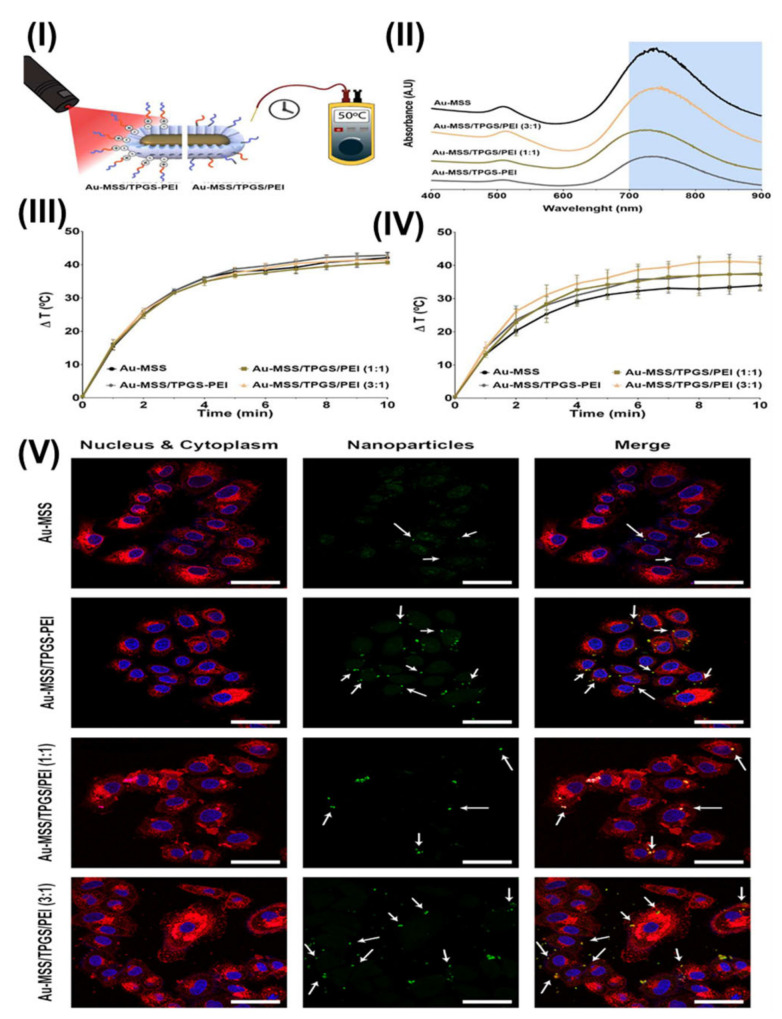
(**I**) Methodology flowchart for determining the in vitro PTT capability of Au-MSS formulations; (**II**) Nanorod UV-vis spectra of Au-MSS and its analogs; (**III**) Complex medium temperature-dependence of the Au-MSS derivatives; (**IV**) The behavior of Au-MSS and its derivatives in ultrapure water as a function of temperature and (**V**) Images of HeLa cells ingesting Au-MSS formulations, as observed by confocal microscopy. Reproduced with permission from the Ref. [134], Figures 4 and 7. (Elsevier^©^ 2019).

**Table 1 pharmaceutics-15-00722-t001:** Pharmacokinetic outcomes of several docetaxel-loaded TPGS-modified nanomedicines administered intravenously at the concentration of 7.5 mg/kg.

TPGS-Modified Nanomedicines	In-Vivo Pharmacokinetic Studies	Outcomes
Trastuzumab targeted TPGS-Chitosan nanoparticles [66]	Animal model: Charles Foster (CF) rats	AUC and MRT of targeted formulation were improved, 2.82-times enhancement in relative and achieved 5.94 folds longer half-life in comparison to Docel™
Transferrin receptor targeted TPGS bioadhesive micelles [67]	Animal model: Charles Foster (CF) rats	AUC and MRT of targeted formulation were improved by 4.08 and 14 times, respectively.
Transferrin receptor targeted TPGS-chitosan cross-linked Chitosan nanoparticles [68]	Animal model: Charles Foster (CF) rats	AUC and MRT of targeted formulation were improved by 4.10 and 10 times, respectively.
Transferrin conjugated TPGS micelles for brain cancer therapy [69]	Animal model: Male Sprague–Dawley rats	After 0.5 h of treatment, it is 18.68 times more efficient than the docetaxel control, and after 2 h of treatment, it is 23.09 times more effective.

**Table 2 pharmaceutics-15-00722-t002:** Summary of TPGS-based nanoformulations.

Authors	Nanotherapeutics	Synthesis Method	Drug	Potential Outcomes	Ref.
Muthu et al., 2015	Micelles	Solvent casting method	Docetaxel	Concurrent transport of DTX and AuNC for concurrent tumor theranostics using TPGS coupled with transferrin. The IC_50_ data revealed that non-targeted micelles and targeting moiety functionalized micelles were 15.31 and 71.73-fold more potent than Taxotere^®^ after 24 h of incubation with MDA-MB-231-luc cells.	[86]
Sonali et al., 2015	Micelles	Solvent casting method	Docetaxel	Non-targeted micelles were <20 nm in size, whereas targeted micelles were achieved at the same size as non-targeted. Micelles were found to have 85% of the drug content. Due to the nanomaterial-based drug delivery, and solubility and penetration improvements that enabled better and prolonged brain localization of targeted micelles than DTX, non-targeted micelles, and the Docel^®^, it was demonstrated that transferrin-functionalized TPGS micelles might be a suitable vehicle for brain targeted drug delivery.	[69]
Muthu et al., 2012	Micelles	Solvent exchange method	Docetaxel	DTX-encapsulated TPGS micelles had a particle size 12–14 nm. DTX loaded in TPGS micelles of high, intermediate, and lower drug-encapsulation ratios had minimal IC_50_ values than the marketed formulation after 24 h of treatment with C6 glioma brain tumor cells.	[87]
Kutty et al., 2015	Micelles	Solvent casting method	Docetaxel	Targeting micelles delivered docetaxel to tumors well, as evidenced by the higher level of tumor growth inhibition observed with docetaxel compared with Taxotere^®^ 15 days following therapy. Inhibition of the blood vessel formation and prevention of metastasis effects were also seen with the micellar formulations, both targeted and non-targeted formulations. The micellar preparation efficiently targeted and suppressed MDA-MB-231 tumors in cell line studies.	[88]
Raju et al., 2013	Liposomes	Solvent injection method	Docetaxel	TPGS liposomes conjugated with trastuzumab for targeted and controlled delivery of docetaxel. After 24 h of treatment with SK-BR-3 cells, the IC_50_ values for the commercially available forms of docetaxel, TPGS liposomes, and trastuzumab functionalized liposomes. TPGS liposomes were determined to be 0.23 ± 1.95, 3.74 ± 0.98, 0.08 ± 0.4 μg/mL, respectively. Trastuzumab-conjugated liposomes had a half-life of 1.9 and 10 times longer than PEG-coated liposomes and DTX, respectively, in in vivo PK studies. An increase of 3.47 and 1.728 times was seen in the AUC.	[89]
Muthu et al., 2012	Liposomes	Solvent injection method	Docetaxel	Liposomes with an average diameter of 202 and 210 nanometers were determined to be non-targeting and targeting, respectively. For Taxotere^®^, non-targeted liposomes, and targeted liposomes, the IC_50_ values were 9.54 ± 0.76, 1.56 ± 0.19 and 0.23 ± 0.05 μg/mL, respectively, after 24 h of treated with MCF-7 cells. Because they were more effective than non-targeted liposomes, the targeted theranostic liposomes have significant promise for improving cancer imaging and therapy.	[9]
Muthu et al., 2011	Liposomes	Solvent injection method	Docetaxel	The coated liposomes varied from 126 to 191 nm in particle size. An electron microscope capable of high-resolution field-emission transmission imaging verified that the liposomes had been coated with TPGS. For the commercialized Taxotere^®^, the naked, the PEG-wrapped, and the TPGS-encased liposomes, it was found that the IC50 value, or the drug level required to destroy 50% of cells in 24 h, was 37.04 1.05, 31.04 0.75, 7.70 0.22, and 5.93 0.57 g/mL, respectively.	[78]
Sonali et al., 2016	Liposomes	Solvent injection method	Docetaxel	Non-targeted and targeted liposome particle sizes were determined to range between 100 and 200 nm in diameter. By using liposomes, almost 70% of drug encapsulation efficiency was obtained. RGD-TPGS-decorated liposomes maintained 80 percent of their drug release for more than 72 h. Following two and four hours of treatment, the results showed that RGD-TPGS-coated multifunctional liposomes outperformed Docel^®^ 6.47- and 6.98-fold, respectively.	[90]
Agarwal et al., 2017	Polymeric nanoparticles	Solvent evaporation method	Docetaxel	Developed formulations had IC_50_ values that were 27 and 148 times greater than Docel^®^ in cytotoxic assays. The non-targeted nanoformulation had 3.23-fold and the targeted nanoformulation had 4.10 fold greater bioavailability than Docel^®^ in an in vivo pharmacokinetic analysis.	[68]
Mehata et al., 2019	Chitosan-based nanoparticles	Emulsification, solvent evaporation, and ionic gelation method	Docetaxel	Non-targeted, conventional nanoparticles had an entrapment efficacy of 74%, whereas targeted nanoparticles had a size of 126–186 nm. The uptake and cytotoxicity of docetaxel-loaded TPGS-g-chitosan nanoparticles with a potential bioadhesion characteristic were improved in vitro experiments on SK-BR-3 cells when compared to Docel^®^, a traditional formulation.	[66]
Vijayakumar et al., 2016	Polymeric nanoparticles	Film hydration method	Resveratrol	To improve resveratrol’s circulation time, biological half-life, and brain passive targeting using TPGS, the study was carried out to conduct this research. There was no burst release with liposomal formulations for up to 48 h of continuous release. The cytotoxicity of RSV-TPGS-Lipo was substantially greater than that of RSV and RSV-Lipo. The intravenous delivery of RSV, RSV-Lipo, and RSV-TPGS-Lipo was found to be the safest.	[91]
Vikas et al., 2021	Chitosan nanoparticles	Emulsification, solvent evaporation, and ionic gelation method	Docetaxel	Chitosan nanoparticles containing docetaxel were produced and various characterization were performed. Chitosan nanoparticles with dual-targeted were tested in vitro and shown to be more cytotoxic to A-549 cells. In comparison to DXL control, the IC_50_ value of the targeted chitosan nanoparticles functionalized with two targeting moieties was about 34 times lower, whereas Wistar rats used in in vivo pharmacokinetic research showed greater relative bioavailability of all NP than did DXL-treated animals.	[60]
Vikas et al., 2022	Chitosan-algiante nanoparticles	Ionic gelation technique	Cabazitaxel	Compared to non-targeted and single-receptor targeted nanoparticles, the absorption of dual-receptor targeted nanoparticles by A-549 cells was much higher in a cellular uptake investigation. The IC_50_ values for Cabazitaxel-loaded dual receptor-targeted nanoparticles against A-549 cells were similarly much lower (38 times) compared to the Cabazitaxel control. The safety of dual receptor-targeted nanoparticles has also been established in vivo through histopathological studies in Wistar rats.	[92]
Vijayakumar et al., 2016	Polymeric nanoparticles	Single-emulsion solvent-evaporation technique	Resveratrol	In this work, the primary goal was to increase the RSV’s systemic circulation time and its biological half-life by utilizing nanoparticles containing PLGA-TPGS-NPs. Anticancer and hematological properties of RSV-PLGA-TPGS-NPs were demonstrated by testing their cytotoxicity against C6 cells, cellular internalization, and haemocompatibility. RSV-PLGA-TPGS-NPs had a plasma half-life of about 18.11 times longer than RSV solution, according to pharmacokinetic investigations. RSV-PLGA-TPGS NPs accumulated more in the brain than RSV in tissue distribution tests.	[10]
Singh et al., 2016	Carbon nanotubes	Nano-extraction method	Docetaxel	Carbon nanotubes (CNT’s) conjugated with TPGS and encapsulated with DTX as a model medication for successful lung tumor therapy. After 24 h of treatment with A549 cells, the IC_50_ values showed that the TPGS-conjugated CNT was 80 times more potent than Docel^®.^ CNT-TPGS conjugated showed malignant cells (*p* < 0.05) in the sub G1 phase of the cell cycle.	[93]
Viswanadh et al., 2021	Polymeric nanoparticles	Dialysis method	Docetaxel	Cetuximab functionalized redox-sensitive TPGS-SH nanoparticles developed with particle size in the range of 183 nm to 227 nm. At pH 5.5, GSH 20 mM, drug release of 94.5 percent was found within 24 h of in vitro release tests in a medium containing varied doses of GSH. The pH and redox sensitivity experiments showed that NPs were more stable at higher pH and lower GSH concentrations. A549 cells were used to perform cytotoxicity, uptake, migration, and apoptosis experiments, and the results showed that the targeted formulation was more hazardous (with a lower IC_50_ value) and hindered the migration of the cells. TPGS-SH NP was shown to have a greater effect on cell death in the benzo(a)pyrene-induced lung cancer model when tested in vivo.	[61]

**Table 3 pharmaceutics-15-00722-t003:** Clinical trials of TPGS and its patents.

Clinical Trials of TPGS		
Title	Phase, Status	NCT Number
Riboflavin and vitamin E expedite topo-pachimetric Epi-on cross-linking significantly more than the Dresden protocol	Not Applicable, Completed	NCT05019768
Evaluation of the OZ439 + TPGS Formulation of Oral Piperaquine in the Fasted State in Healthy Volunteers	Phase 1, Completed	NCT01853475
Piperaquine and OZ439: An Open-Labeled Pharmacokinetic Study of OZ439+TPGS Granules for Oral Suspension Given Alone or in Combination with Piperaquine Phosphate Tablets or Granules for Oral Solution in Healthy Volunteers	Phase 1, Completed	NCT01958619
Berberine Absorption in Humans, a Bioactive Compound	Not Applicable, Completed	NCT03438292
DSM265’s Effect on Malaria and Food Bioavailability	Phase 1, Completed	NCT03637517
Summary of TPGS-based materials		
Title	Inventor Name	Patent No
Vitamin E TPGS concentrated fluid with a low water content	Jandzinski et al. [137]	US20060O88558A1
Vitamin E TPGS molecules, which are present in pharmaceutical formulations, have been shown to facilitate the metabolism of lipophilic drugs without causing a noticeable decrease in the drug’s efficacy.	Hyatt et al. [138]	WO2006039268A3
A new class of surfactant-like materials comprising vitamin E TPGS and water-soluble polymer	Li et al. [135]	EP 1 799 194 B1
Innovative viscoelastic material based on alginate and TPGS/TPGSA	Xia et al. [139]	EP 1 796 628 B1
TPGS-750-M-Containing Compositions	Volker Berl [140]	US20160340332A1
Consistently effective pharmaceutical formulations	Durak et al. [141]	US 2015O141351A1
Poorly soluble medications are transported in emulsion carriers	Lambert et al. [142]	US 2003OO27858A1
Effective bioavailability of lipophilic substances using solid coprecipitates	Amselem et al. [143]	USOO5891469A
Assemblies and synthesis of surfactants	Volker Berl [144]	US 2011 O184194A1
Efficient Drug Delivery Through Nanoparticle Coating	Si-Shen Feng [145]	US 2006O1885.43A1

## Data Availability

Not applicable.

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
