# Peer review of "Vitamin E TPGS-Based Nanomedicine, Nanotheranostics, and Targeted Drug Delivery: Past, Present, and Future"

_pharmaceutics, 2023, doi:10.3390/pharmaceutics15030722_

Round 1

Reviewer 1 Report

Adnan M. Jasim and Mohammed J. Jawad in 2021 already published chapter Pharmaceutical Applications of Vitamin E TPGS

DOI: 10.5772/intechopen.97474

It would be appropriate to discuss this publication in Introduction and demonstrate the difference of your review compared with previously published reviews on the similar topics.

It would be beneficial to more clearly identify the PAST of TPGS

It has been reported in lines 163 - 168 that "to produce TPGS, ɑ-tocopheryl succinate (ɑ-TOS) is esterified with PEG1000. This  amphiphilic agent's favorable HLB value makes it effective as a solubilizer, emulsifier,  penetration, and absorption enhancer of hydrophobic medications. Increased medication efficacy against multidrug-resistant cells, decreased particle size, and greater solubility  are all possible outcomes of TPGS-formed micelles in an aqueous solution. By inhibiting P-gp expression and modifying efflux pump activity, TPGS can reduce MDR" and it is the PRESENCE of TPGS.

The abbreviation MDR has not been deciphered  in the text.

And it would be especially beneficial to much more clearly and specifically to identify the FUTURE of TPGS

A number of relevant recent publications have not been cited, for example

Advances in understanding the role of P-gp in doxorubicin resistance: Molecular pathways, therapeutic strategies, and prospects, Drug Discovery Today, 10.1016/j.drudis.2021.09.02027, 2, (436-455), (2022)

Nanomedicines for combating multidrug resistance of cancer, WIREs Nanomedicine and Nanobiotechnology, 10.1002/wnan.171513, 5, (2021).

Progress in the study of D-α-tocopherol polyethylene glycol 1000 succinate (TPGS) reversing multidrug resistance, Colloids and Surfaces B: Biointerfaces, 10.1016/j.colsurfb.2021.111914205, (111914), (2021).

Vitamin E-tocopheryl polyethylene glycol succinate decorated drug delivery system with synergistic antitumor effects to reverse drug resistance and immunosuppression, Colloids and Surfaces A: Physicochemical and Engineering Aspects, 10.1016/j.colsurfa.2021.127387628, (127387), (2021).

Yuan Z, Yuan Y, Han L, et al. Bufalin-loaded vitamin E succinate-grafted-chitosan oligosaccharide/RGD conjugated TPGS mixed micelles demonstrated improved antitumor activity against drug-resistant colon cancer. Int J Nanomedicine. 2018;13:7533-7548. doi: 10.2147/IJN.S170692.

Mohyeldin et al. Hybrid lipid core chitosan-TPGS shell nanocomposites as a promising integrated nanoplatform for enhanced oral delivery of sulpiride in depressive disorder therapy. Int J Biol Macromol, 2021, 188,432-449

Bi, Fengyu, et al. "Development of antioxidant and antimicrobial packaging films based on chitosan, D-α-tocopheryl polyethylene glycol 1000 succinate and silicon dioxide nanoparticles." Food Packaging and Shelf Life 24 (2020): 100503.

Yong, Huimin, et al. "Preparation and characterization of antioxidant packaging by chitosan, D-α-tocopheryl polyethylene glycol 1000 succinate and baicalein." International journal of biological macromolecules 153 (2020): 836-845.

Sampath, Malathi, et al. "The remarkable role of emulsifier and chitosan, dextran and PEG as capping agents in the enhanced delivery of curcumin by nanoparticles in breast cancer cells." International Journal of Biological Macromolecules 162 (2020): 748-761.

Mehata, Abhishesh Kumar, et al. "Chitosan-alginate nanoparticles of cabazitaxel: Design, dual-receptor targeting and efficacy in lung cancer model." International Journal of Biological Macromolecules 221 (2022): 874-890.

Guo T et al. TPGS assists the percutaneous administration of curcumin and glycyrrhetinic acid coloaded functionalized ethosomes for the synergistic treatment of psoriasis. Int J Pharm. 2021 Jul 15;604:120762. doi: 10.1016/j.ijpharm.2021.120762.

https://doi.org/10.3389/fphar.2019.00769

Liu, Z et al. Increasing Cellular Uptake and Permeation of Curcumin Using a Novel Polymer-Surfactant Formulation. Biomolecules 2022, 12, 1739. https://doi.org/10.3390/ biom12121739

  1.  Zhang H et al. (2020) Enhanced oral bioavailability of self-assembling curcumin–vitamin E prodrug-nanoparticles by co-nanoprecipitation with vitamin E TPGS, Drug Development and Industrial Pharmacy, 46:11, 1800-1808, DOI: 10.1080/03639045.2020.1821049
  2.  
  3. Sharifi-Rad J et al. (2021) Resveratrol-Based Nanoformulations as an Emerging Therapeutic Strategy for Cancer. Front. Mol. Biosci. 8:649395. doi: 10.3389/fmolb.2021.649395

Gregoriou Y et al. Resveratrol loaded polymeric micelles for theranostic targeting of breast cancer cells. Nanotheranostics 2021; 5(1):113-124. doi:10.7150/ntno.51955. https://www.ntno.org/v05p0113.htm

Pacl HT et al. Water-soluble tocopherol derivatives inhibit SARS-CoV-2 RNA-dependent RNA polymerase. bioRxiv [Preprint]. 2021 Jul 27:2021.07.13.449251. doi: 10.1101/2021.07.13.449251.

  1. So the additional review of above relevant publications could be helpful especially for the determination of FUTURE trends in the topic.

  2. The addition of information about dependence of MDR on substrate stiffness and tissue integrity would be an advantage.

  3. The extension of review will allow to make additional conclusions related with future perspectives.

Reviewer 2 Report

In this paper, the authors summarized the recent advances about the applications of vitamin E TPGS in nanomedicine and nanotheranostics. The topic in this area is important and timely, so a review emphasizes recent progress is useful and valuable.

1.     In “2. Historical milestone achieved in TPGS developments”, the authors tried to highlight some significant development of TPGS, it could be better to arrange the progress in chronological order in a figure or scheme.

2.     The quality of some pictures in figures 8 and 11 should be improved because some characters are illegible.

3.     Line 190 “I It was suggested that liposomes…” should be “It was suggested that liposomes…”

4.     Several abbreviations in the figures should be explained, such as the “TIG” and “TCT” in figure 12.
